# Crosstalk between MSH2–MSH3 and pol β promotes trinucleotide repeat expansion during base excision repair

Yanhao Lai[1], Helen Budworth[2], Jill M. Beaver[3], Nelson L.S. Chan[2], Zunzhen Zhang[4], Cynthia T. McMurray[2] & Yuan Liu[1,3,5]

Studies in knockout mice provide evidence that MSH2–MSH3 and the BER machinery promote trinucleotide repeat (TNR) expansion, yet how these two different repair pathways cause the mutation is unknown. Here we report the first molecular crosstalk mechanism, in which MSH2–MSH3 is used as a component of the BER machinery to cause expansion. On its own, pol β fails to copy TNRs during DNA synthesis, and bypasses them on the template strand to cause deletion. Remarkably, MSH2–MSH3 not only stimulates pol β to copy through the repeats but also enhances formation of the flap precursor for expansion. Our results provide direct evidence that MMR and BER, operating together, form a novel hybrid pathway that changes the outcome of TNR instability from deletion to expansion during the removal of oxidized bases. We propose that cells implement crosstalk strategies and share machinery when a canonical pathway is ineffective in removing a difficult lesion.

[1] Department of Chemistry and Biochemistry, Florida International University, 11200 SW 8th Street, Miami, Florida 33199, USA. [2] Life Sciences Division, Lawrence Berkeley National Laboratory, 1 Cyclotron Road, 33R249, Berkeley, California 94720, USA. [3] Biochemistry Ph.D. Program, Florida International University, 11200 SW 8th Street, Miami, Florida 33199, USA. [4] Department of Occupational and Environmental Health, Sichuan University West China School of Public Health, 16#, Section 3, Renmin Nan Lu, Chengdu, Sichuan 610041, China. [5] Biomolecular Sciences Institute, School of Integrated Sciences and Humanity, Florida International University, 11200 SW 8th Street, Miami, Florida 33199, USA. Correspondence and requests for materials should be addressed to Y.Liu (email: yualiu@fiu.edu) or to C.T.M. (email: ctmcmurray@lbl.gov).

Mammalian cells have evolved sophisticated DNA repair systems to correct mispaired or damaged bases and extrahelical loops. In the mismatch repair (MMR) pathway, MutS-like heterodimers (MutSα, MSH2–MSH6 and MutSβ, MSH2–MSH3) recognize chemically modified or extrahelical DNA, and initiate sequential assembly of downstream repair machinery to repair the lesion[1–3]. Surprisingly, however, the eukaryotic mismatch recognition complex, MSH2–MSH3, not only fails to act as a guardian of the genome at the long disease-length trinucleotide repeat (TNR) tracts but also causes expansion, the lethal mutation underlying Huntington's disease (HD) as well as at least 30 other fatal diseases[4–8].

Indeed, crossing of mice harbouring expandable triplet repeats with mice lacking the MMR protein MutS homologue 2 (MSH2) or MutS homologue 3 (MSH3) attenuates expansion in the 5′-CAG-3′ repeats in the human HD gene[9–16], the 5′-CTG-3′ repeats in the 3′-untranslated region of the human myotonic dystrophy 1 protein kinase transgene[17–20], the 5′-GAA-3′ repeats in the FXN gene in Friedreich's ataxia (FRDA)[21] and the 5′-CGG-3′ repeats in the fragile mental retardation gene in fragile X syndrome (FXS)[22]. Loss of Pms2 (ref. 23), and other MutL homolog (MLH)-related endonucleases[24,25], also suppresses expansion, bolstering the notion that the MMR pathway contributes to expansion. Loss of MSH6 has little, tissue-specific, or even protective effects on expanding TNRs in most animal models[11,14,17,19]. Indeed, human cell line experiments agree with the fact that MSH3 is the causative agent, and MSH6 is less important in this process[26]. These unexpected results provided genetic evidence that MSH2–MSH3 causes, rather than corrects, the expansion mutation. A causative role for MSH2–MSH3 has been confirmed in multiple cell models for disease[27–29].

MSH2–MSH3 binds well to small loops and mispaired bases in hydrogen bonded TNR loops[11,30–33]. Although there is general agreement that expansion arises from faulty processing of non-B form DNA, the role of MSH2–MSH3 in this process remains enigmatic. Normally, in post-replicative repair, MSH2–MSH3 initiates successful removal of small loops by 5′–3′ exonuclease activity and does so without mutation. However, expansion in non-dividing cells requires an unrepaired loop, whose integration into duplex DNA increases the length of the TNR tract. Thus, MSH2–MSH3 can cause expansion by either facilitating loop formation, failing to remove the loop, or by aiding loop incorporation into duplex DNA. Although the results from diverse model organisms imply that other DNA repair pathways can promote TNR expansion, the preponderance of evidence indicates that the MMR system remains arguably the most important.

A mechanistic role for the MMR pathway, however, has been difficult to establish in vitro. Much attention has been paid to somatic expansion, whose suppression delays disease onset[34,35], yet in vitro mechanistic analyses support conflicting models. For example, irreversible dissociation of MSH2–MSH3 from the hairpin could block its removal, yet biochemical analysis confirms that MSH2–MSH3 binds equally well and in a modest range to CAG/CTG hairpins and to repair competent, small, unstructured loops, which are good substrates for MMR-dependent removal[11,30–32]. Moreover, plasmids harbouring repeat tracts undergo small insertions and deletions in cell extracts lacking MSH2, implying that it is not needed for loop repair[36]. In at least one other analysis, CAG hairpins are removed efficiently in vitro by a process that resembles MMR[37]. Despite clear genetic data, it remains highly controversial how MSH2–MSH3 causes expansion.

Part of the difficulty may be the multi-functional nature of MSH2–MSH3, which has emerged as a component in double-strand break repair[38,39], transcription-coupled repair (TCR)[40–43]

and base excision repair (BER). Indeed, at TNRs, expansion is reduced in transgenic mice lacking 8-oxo-guanine glycosylase (OGG1)[44], NEIL1 (ref. 45), pol β with normal activity[46] and XPA[47], strongly supporting the notion that excision repair machinery also contributes to expansion during the removal of oxidative damage in vivo. Consistent with such a role, oxidative DNA-damaging agents promote instability in CTG repeats in human embryonic kidney cells[48], CAG repeats in HD cells[49], CGG repeats in FXS transgenic animals[50] and CAG repeats in human HD fibroblasts[44].

As with the MMR pathway, however, no clear mechanisms are obvious. XPA is part of the NER pathway, which has two subpathways, one for global genome repair and the other for transcribed genes (TCR)[51]. In mice, the loss of XPC, the recognition protein for global genome repair, has no effect on instability in an HD mouse model[14]. Cockayne's syndrome B protein (CSB), a DNA remodelling initiator of TCR, stabilizes repeats at least in FXS[52] and HD[40] mouse models. Loss of BER machinery (OGG1, NEIL1 and an inactive pol β mutant) suppresses expansion in mice. However, in vitro, lesion repair by BER causes substantial deletions of TNRs in systems reconstituted using purified components[53–55].

Despite years of investigation, there remain puzzling disconnects between the in vivo genetic predictions and in vitro mechanistic testing of TNR dynamics at disease-length alleles. Multiple pathways appear to contribute to expansion, but how or under what conditions they are used remains poorly understood. Here to the best of our knowledge, we provide the first direct evidence for a molecular crosstalk mechanism, in which MSH2–MSH3 is used as a component of the BER machinery to cause expansion. MSH2–MSH3 forms a physical complex with pol β, and when operating in the context of BER, stimulates pol β synthesis through the TNRs, switching the outcome of BER from TNR deletion to TNR expansion. The MMR and BER, operating together, form a novel hybrid pathway that coordinates the activities of more than one pathway to both suppress deletion and promote expansion of TNR tracts during the removal of oxidized bases.

## Results

**MSH2–MSH3 and pol β physically interact at TNR tracts.** BER comprises a series of well-characterized steps, which lead to removal of a damaged base and restoration of an intact duplex (Supplementary Fig. 1). An abasic site forms as a BER intermediate after the damaged base is removed by a DNA glycosylase, and the 5′-end is processed by apyrimidinic endonuclease 1 (APE1) forming a gap that is filled by a DNA polymerase, often polymerase β (pol β; Supplementary Fig. 1). Since MSH2–MSH3, OGG1, NEIL1 and pol β have been implicated in causing expansions in mice, we asked whether there are interactions between the MMR and BER machinery.

There was little evidence for a direct interaction of MSH2–MSH3 with the lesion recognition glycosylase, OGG1, on CAG templates harbouring an 8-oxo-G template (Supplementary Fig. 2; Supplementary Method). Thus, we assessed the interaction of MSH2–MSH3 with the downstream BER machinery on a synthetic BER template, which mimicked an abasic site after glycosylase removal of the oxidized base. The BER templates comprised a $(GAA)_{20}$ or $(CAG)_{20}$ repeat tract (Fig. 1a, red) flanked on either side with 20 bases of random sequences (Fig. 1a, black). The abasic site mimic, tetrahydrofuran (THF), was substituted for the guanine of the tenth GAA or CAG unit, and, in this case, leaves a widowed C that lacks a complementary nucleotide within an intact duplex (Fig. 1a). The THF residue represented a scenario where the lesion resided in the centre of the TNR tract, and divided it into 9 TNRs on the 5′-side and 10 TNRs on the 3′-side of the

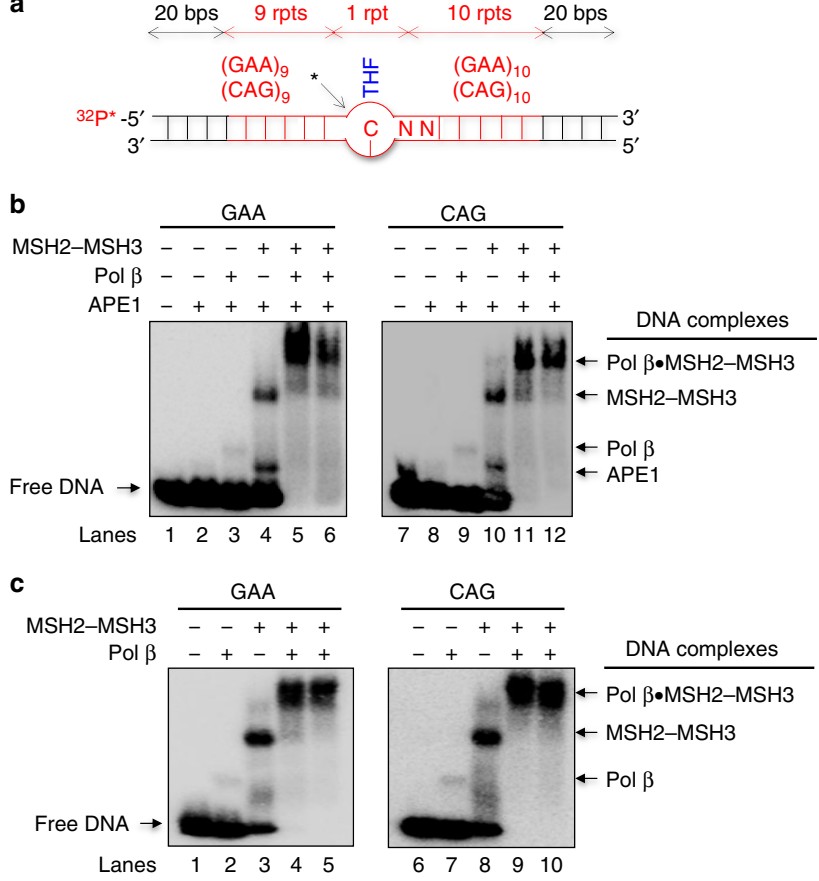

**Figure 1 | MSH2–MSH3 forms a physical complex with pol β that loads it onto the TNR tract at the APE1 incision site.** (**a**) A schematic representation of the synthetic TNR templates used in the BER reactions. Each comprised 100 bases with a total of 20 GAA or CAG TNRs (in red) and 20 bases of random sequence on either end represented in black. One repeat modified with a tetrahydrofuran (THF, in blue) separates the 20 repeats, with nine TNRs on the left side and 10 TNRs on the right side. The THF site mimics the abasic site intermediate for BER, and is cleavable by APE1 (black arrow and star) to initiate the pol β extension reaction. (**b**,**c**) The formation of a pol β•MSH2–MSH3•DNA ternary complex was detected on $(GAA)_{20}$ or $(CAG)_{20}$ repeat-containing substrates by gel mobility shift assay as described in the Methods section. The composition of each reaction is indicated at the top of the gel: ( + ) is the presence and ( − ) is the absence of the component. Lanes 1 and 7 (**b**), and lanes 1 and 6 (**c**) correspond to the substrate only. Lanes 2 and 8 (**b**) correspond to binding mixture with 25 nM APE1. Lanes 3 and 9 (**b**), and lanes 2 and 7 (**c**) correspond to binding mixture with 10 nM pol β. Lanes 4 and 10 (**b**), and lanes 3 and 8 (**c**) correspond to binding mixture with 100 nM MSH2–MSH3. Lanes 5 and 11 (**b**), and lanes 4 and 9 (**c**) correspond to binding mixture with 10 nM pol β in the presence of 100 nM MSH2–MSH3, where pol β was added together with MSH2–MSH3. Lanes 6 and 12 (**b**), and lanes 5 and 10 (**c**) correspond to binding mixture with 10 nM pol β in the presence of 100 nM MSH2–MSH3, where MSH2–MSH3 was incubated with the substrates before the addition of pol β. The complexes are indicated to the right of the gel. An unidentified band migrating between APE1•DNA and pol β•DNA complexes in **b** (lanes 4 and 10) and (**c**) (lanes 2 and 8) most likely represents a complex of DNA with a small truncation of MSH2–MSH3. The experiments were repeated at least three times. rpt, repeat.

lesion (Fig. 1a). We performed three complementary experiments to test for physical interactions among components.

In the first experiment, we tested whether MSH2–MSH3 would bind to a lesion classically repaired by BER. Purified components of the BER machinery were incubated with the GAA or CAG synthetic templates, which were $^{32}$P-labelled on the 5′-end of the damaged strand (Fig. 1b). To initiate BER, the abasic site (THF site) in the substrate was cleaved on its 5′-side using 25 nM APE1, which created a 3′-OH for polymerase extension (Fig. 1b). The reaction was cooled before the addition of pol β to prevent further enzymatic activity. Agarose polyacrylamide gels resolved the bound complexes by band shift[56].

Pol β and MSH2–MSH3 independently formed binary complexes with both $(GAA)_{20}$ or $(CAG)_{20}$ substrates (Fig. 1b, lanes 3–4 and 9–10; pol β•DNA and MSH2–MSH3•DNA). MSH2–MSH3 and pol β alone were independently capable of binding to the 1-nt gap intermediate that was generated by APE1 5′-incision of the abasic site (Fig. 1b, lanes 3–4 and 9–10; Fig. 1c,

lanes 2–3 and 7–8). The binding of MSH2–MSH3 was far greater than that of pol β despite the fact that the latter is a classic component of BER (Fig. 1b, lanes 3–4 and 9–10; Fig. 1c, lanes 2–3 and 7–8). The addition of pol β and MSH2–MSH3 together, however, led to a striking and synergistic formation of a protein–DNA 'super-shifted' band (Fig. 1b, lanes 5–6 and 11–12; Fig. 1c, lanes 4–5 and 9–10). The super-shifted band was observed whether pol β was added together with MSH2–MSH3 and DNA substrates (Fig. 1b, lanes 5 and 11; Fig. 1c, lanes 4 and 9) or was added last (Fig. 1b, lanes 6 and 12; Fig. 1c, lanes 5 and 10). The formation of a pol β•MSH2–MSH3•DNA complex did not depend on APE1 (Fig. 1c). Rather, the pol β•MSH2–MSH3•DNA complex formed a super-shifted band in the absence of APE1, on both $(GAA)_{20}$ and $(CAG)_{20}$ substrates that mimicked the APE1 precut intermediates (Fig. 1c, lanes 4–5 and 9–10). Thus, MSH2–MSH3 acted as part of the BER machinery at the APE1 endonucleolytic nick site, the initiating intermediate for BER.

DNase I footprinting assay confirmed that binding sites for pol β and MSH2–MSH3 partially overlapped on both the $(GAA)_{20}$ (Fig. 2a–c) and $(CAG)_{20}$ (Fig. 2d–f) repeat substrates. Indeed, when labelled on the 5′-end of the $(GAA)_{20}$ or $(CAG)_{20}$ template strand (Fig. 2a,d), the footprint of both pol β and MSH2–MSH3 proteins overlapped near the THF (Fig. 2c,f). Pol β had a bi-partite footprint, which straddled the THF damage at position $+53$ (Fig. 2a, lanes 3–4). MSH2–MSH3 binding overlapped with that of pol β, but also protected a larger region (Fig. 2a,d, lanes 5–6). No footprint was observed in the absence or presence of DNase I without protein (Fig. 2a,d, lanes 1–2). Interestingly, binding of both proteins was asymmetric. There was little protection from either protein on the damaged strand (top strand) of the $(GAA)_{20}$ and $(CAG)_{20}$ substrates (Fig. 2b,e). However, protection by both proteins was detected on the template strand (bottom strand) (Supplementary Fig. 3). Thus, pol β and MSH2–MSH3 shared binding sites on the template strand of the substrates at the APE1 nicked site, and the complex was positioned with MSH2–MSH3 at or slightly ahead of pol β (Fig. 2c,f).

**Formation of the pol β•MSH2–MSH3 complex in cells**. If a physical complex between pol β and MSH2–MSH3 was relevant to TNR expansion by a BER-dependent mechanism, we expected that the two proteins would directly interact in cells, and should be simultaneously recruited to lesions produced by oxidative damage. Thus, in a second set of experiments, we performed immunoprecipitation (IP) and co-localization (Fig. 3) to test whether pol β and MSH2 formed a stable complex in cells, and did so on TNR tracts. Tested was complex formation before and after treatment with oxidative DNA-damaging agents in normal lymphoblasts or in lymphoblasts from a FRDA patient (GAA expansion). Potassium chromate ($K_2CrO_4$) or potassium bromate ($KBrO_3$) was used as DNA-damaging agents, since the agents increase the level of oxidized bases that would require removal by BER.

Indeed, pol β and MSH2–MSH3 formed a physical complex in cells as judged by IP (Fig. 3; Supplementary Fig. 4). In both untreated (Fig. 3a) and treated lymphoblasts (Fig. 3b,c), pol β was 'pulled down' in the anti-MSH2 immunoprecipitates (Fig. 3a, IP MSH2, lane 6), and MSH2 was detected in the anti-pol β immunoprecipitates of normal lymphoblasts (Fig. 3a, IP pol β, lane 5). In addition, MSH3 was detected in the anti-pol β and anti-MSH2 immunoprecipitates under all tested conditions (Supplementary Fig. 4, lanes 5–6). Consistent with roles in BER, the IP complex increased in cells treated with oxidizing agents potassium chromate ($K_2CrO_4$) or potassium bromate ($KBrO_3$; Fig. 3b,c; Supplementary Fig. 4b,c, lanes 5–6). In all reactions, the antibodies were specific for each respective protein (Fig. 3a–e; Supplementary Fig. 4, lanes 1 and 2), and the proteins detected by IP were proportional to the input protein (Fig. 3a–c; Supplementary Fig. 4, lane 3). IgG failed to immunoprecipitate pol β, MSH2 or MSH3 from cells (Fig. 3a–c; Supplementary Fig. 4, lane 4), indicating that the IP complexes depended on the antibody. As an additional control, we repeated the IP experiment in LoVo cells, which do not express MSH2 (Fig. 3d). As expected, in those cells, pol β was immunoprecipitated by its own antibody (Fig. 3d, lane 5), but MSH2 was not detected nor did it co-immunoprecipitate with pol β (Fig. 3d, lanes 5 and 6), and vice versa. The interaction was observed in reactions containing the purified proteins (Fig. 3e), ruling out the possibility that a bridging protein mediates the formation of the MSH2–MSH3•pol β protein complex in the cell extracts. Thus, in cells, MSH2–MSH3 and pol β formed a direct physical complex (Fig. 3e), which appeared to increase on oxidative DNA damage.

The results of the co-localization of MSH2–MSH3 and pol β also supported the IP results (Fig. 3f; Supplementary Fig. 5). In normal lymphoblasts, both pol β and MSH2 are constitutive proteins and exist throughout cells (Fig. 3), and we expected that detection of meaningful interactions would be difficult based on the merged image alone. Therefore, cells untreated or treated with damaging agents were stained with specific antibodies to pol β (Fig. 3f, green) and MSH2 (Fig. 3f, red), and complex formation was evaluated using pixel analysis. That analysis quantifies the number of pixels that contain one, the other, or both emission intensities. Specifically, we imaged optical sections of 1 μm in the middle of each cell, and only those pixels that harboured both red and green signals were considered as meaningful interactions between the proteins (Fig. 3f,g; Supplementary Fig. 5).

We were unable to detect constitutive interactions of pol β and MSH2–MSH3 in untreated lymphoblasts (Fig. 3f, untreated). Most red (Fig. 3f, MSH2) and green (Fig. 3f, pol β) pixels did not overlap (Fig. 3f, Merge), and few pixels contained both red and green intensity (Fig. 3f, pixel analysis). However, DNA-damaging agents induced a marked increase in the number of the pixels that harboured both pol β and MSH2 intensities (Fig. 3f, $K_2CrO_4$ and $KBrO_3$), implying that complexes formed in response to damage. As we observed in the IP experiments, co-localization of MSH2 and pol β was abolished in LoVo cells, which do not express MSH2, and no pixels harboured both red and green signals under these conditions (Fig. 3g). In untreated lymphoblasts from FRDA patients (Supplementary Fig. 5), the number of pixels that harboured both red and green signal intensity was higher relative to untreated normal lymphoblasts (Fig. 3f), and the number of complexes changed little in response to damage (Supplementary Fig. 5). These results are consistent with the fact that oxidative damage is elevated in FRDA cells[57].

We predicted that if the interaction of pol β and MSH2–MSH3 were relevant to expansion, then the pol β•MSH2–MSH3 complex would bind to the TNR regions within the disease genes after damage. Chromatin IP (ChIP) assay was a test of the hypothesis (Fig. 4; Supplementary Fig. 6). No GAA repeat DNA precipitated in the absence of antibody (Fig. 4a, lanes 6, 10 and 14). Only MSH2 co-precipitated with the $(GAA)_{15}$ locus in the *FXN* gene in untreated normal lymphoblasts (Fig. 4a, compare lanes 4 and 5), while both pol β and MSH2 proteins co-precipitated the same site in the treated cells (Fig. 4a, lanes 8–9 and 12–13). The results implied that the complex was recruited to the repeat tract after treatment with either 0.5 mM $K_2CrO_4$ or 10 mM $KBrO_3$. The quantified results confirmed that the enrichment was substantial relative to untreated cells (Fig. 4c). In LoVo cells that lack MSH2, no DNA was detected with MSH2 antibodies (Fig. 4b,d). The GAA repeat sequence recruited pol β only on oxidative DNA damage (Fig. 4a, compare lanes 8 and 12 with lane 4; Fig. 4a,c). Since endogenous DNA damage occurs throughout life in every cell, some pol β was recruited to the CAG repeats in the *HTT* gene in HD patient lymphoblasts independently of exogenous DNA damage (Supplementary Fig. 6a,b), as also shown previously in the striatum of HD mice[58]. Collectively, MSH2–MSH3 and pol β were recruited together to the sites of damage in cells, consistent with a physical cooperation of the two proteins during the removal of oxidized bases by BER.

**MSH2–MSH3 suppresses TNR deletion and promotes expansion**. Since MSH2–MSH3 and pol β bound together at the APE1 nick, we asked whether the interaction played a role in TNR expansion. To test the hypothesis, we pre-cleaved the synthetic BER substrate (Fig. 5a) on the damaged strand at the THF site using 25 nM APE1 to initiate BER, and added to the reaction was pol β, FEN1

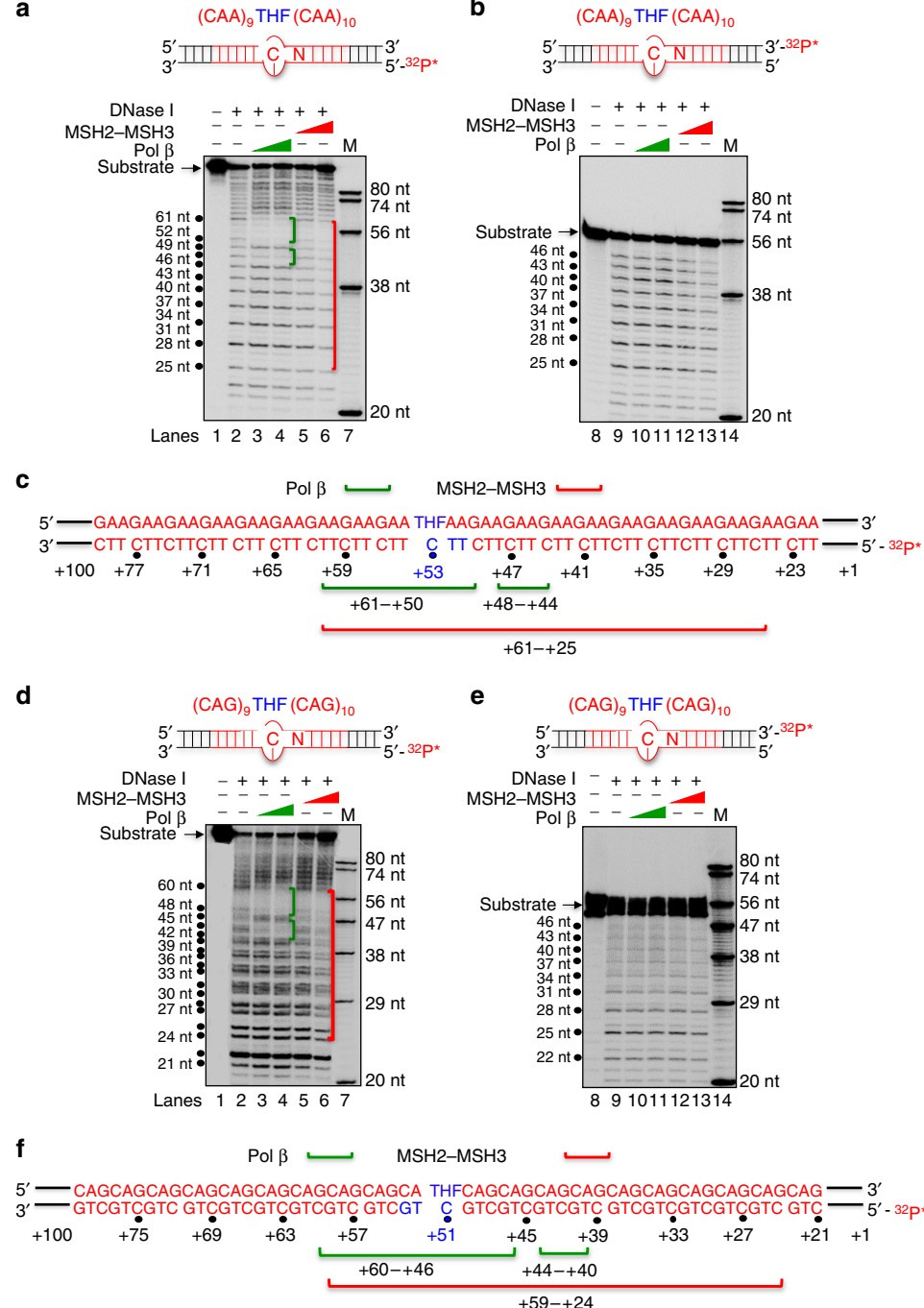

**Figure 2 | MSH2–MSH3 and pol β have overlapping binding sites on TNR substrates preincised with APE1.** DNase I footprinting of pol β and MSH2–MSH3 on (GAA)$_{20}$ (**a,b**) and (CAG)$_{20}$ (**d,e**) substrates. A schematic diagram of the (GAA)$_{20}$ template depicting the position of the THF site ( + 53 in blue) and the $^{32}$P-label on the 5′-end of the template strand (**a**) or the 3′-end of the damaged strand (**b**). The binding sites of MSH2–MSH3 and pol β on TNR-containing BER intermediates were probed by DNase I digestion as described in the Methods section. Pol β, MSH2–MSH3 and DNase I (0.008 U) were added as indicated above the gels; ( + ) is the presence of the component and ( − ) is the absence of the component. Lanes 1 and 8 are the substrate only. Lanes 2 and 9 are unbound substrates digested with 0.008 U DNase I. The triangles indicate increasing concentration of either pol β at 50 and 100 nM (in green) or MSH2–MSH3 at 100 and 200 nM (in red). Lanes 7 and 14 are synthesized markers (M), whose size is indicated to the right of the gel. Binding of MSH2–MSH3 and pol β occurs primarily on the template strand (compare **a** and **b**). DNase I digestion sites are indicated to the left of the gel as black dots and the numbered positions of the cleavage products correspond to the numbering system in **c**. (**c**) A summary of the overlapping binding sites of pol β and MSH2–MSH3 on the (GAA)$_{20}$ substrates. The green bracket indicates the footprint of pol β, while the red bracket indicates the footprint of MSH2–MSH3. (**d–f**) Same as **a–c** except for the (CAG)$_{20}$ substrates. The experiments were performed in triplicate.

and ligase I (LIG I) at 37 °C to complete BER without or with 100 nM MSH2–MSH3 (Fig. 5b). The repaired product was separated from the template with avidin beads, and the ligated 5′-FAM (fluorescein amidite) repair fragments were resolved

by high-resolution capillary electrophoresis to define their length (Fig. 5b).

In the absence of MSH2–MSH3, repair of the abasic lesion in the (GAA)$_{20}$ substrate resulted in deletion of seven to nine repeats

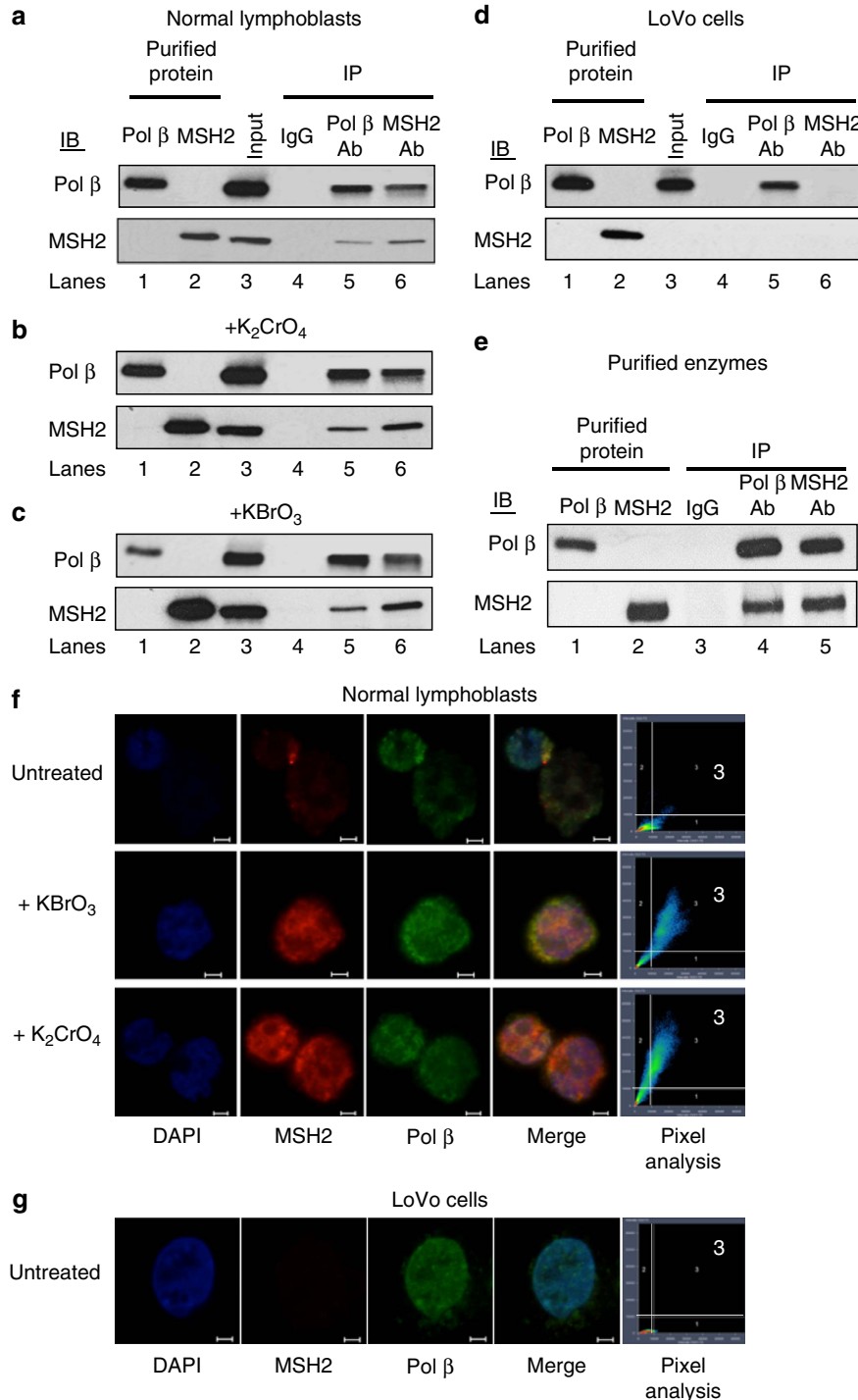

**Figure 3 | MSH2–MSH3 and pol β co-localize in cells to form physical complexes that increase with treatment by DNA-damaging agents.**
Co-immunoprecipitation (IP) and immunoblotting (IB) of MSH2 or pol β in lymphoblast cell extracts derived from a normal individual (GM02152) were either untreated (**a**) or treated with 0.5 mM $K_2CrO_4$ (**b**) or 10 mM $KBrO_3$ (**c**) for 2 h, respectively. (**d**) Same as **a** except for LoVo cells that lack MSH2. Cell lysates were subjected to co-IP and immunoblotting for pol β and MSH2, as described in the Methods section. Lanes 1 and 2 are the purified pol β or MSH2 proteins, as indicated, detected with their respective antibodies. Lane 3 corresponds to cell lysates without treatment as an 'input' control. Lane 4 is the cell lysates immunoprecipitated with a rabbit IgG alone; lanes 5 and 6 are cell lysates immunoprecipitated with an anti-pol β antibody and an anti-MSH2 antibody, respectively. (**e**) Same as **a–d**, but for purified proteins. Purified pol β and MSH2 proteins were employed as a molecular weight marker (lanes 1 and 2). (**f,g**) Co-localization of MSH2 and pol β in normal lymphoblasts (GM02152) (**f**) and LoVo cells (**g**) untreated or treated with oxidative DNA-damaging agents, potassium bromate or potassium chromate, as indicated. Representative images are illustrated. MSH2 (red), pol β (green) or merged images are indicated. Zeiss pixel analysis results are illustrated to the right side of each panel. The x and y axes represent red and green staining intensity, respectively. The white bars represent the quadrant thresholds below which pixels were deemed to contain only red or only green staining. The number 3 is the quadrant in which pixels contain both red and green intensities. 4,6-Diamidino-2-phenylindole (blue) is a stain for nuclear DNA; red is the anti-MSH2 staining; green is anti-pol β staining. (**g**) Same as **f** for untreated LoVo cells. All experiments were carried out in triplicate. All scale bars, 10 μm.

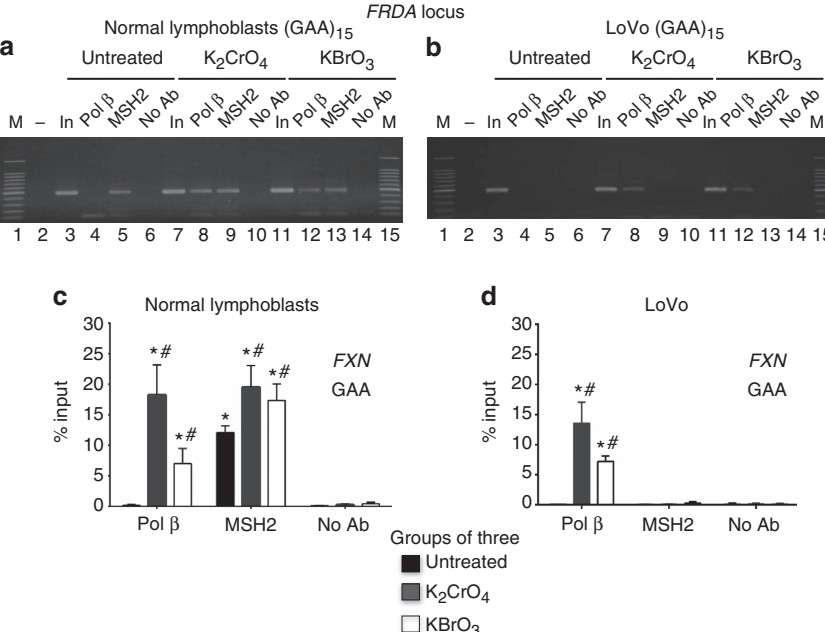

**Figure 4 | Recruitment of MSH2 and pol β to GAA repeats of the *FXN* gene in human lymphoblasts.** Recruitment of MSH2 and pol β to the GAA locus (GAA$_{15}$) in the *FXN* gene in normal lymphoblasts and in LoVo cells. (**a**) GAA repeats in the *FXN* gene of lymphoblasts derived from a normal individual (GM02152) were detected by ChIP assay with an anti-MSH2 and anti-pol β antibody, respectively. ChIP assay was also conducted in LoVo cells as a control. (**a**) Representative agarose gel results showing recruitment of MSH2 and pol β to GAA repeats of the *FXN* gene in normal lymphoblasts after exposure to chromate or bromate or without any treatment. (**b**) Representative agarose gel results showed recruitment of pol β to GAA repeats of the *FXN* gene in untreated LoVo cells or LoVo cells treated with chromate or bromate. Lanes 1 and 15 represent DNA size markers (M). The 'non-template' control, the no antibody control and the input DNA are indicated for both treated and untreated cells. The bands were obtained using quantitative PCR and expressed as '% input'. All '% input' were obtained from three independent experiments and expressed as mean ± s.d. Two-way analysis of variance with Tukey's multiple comparison post tests was used to determine statistically significant differences. *$P < 0.05$, compared with No-Ab control and #$P < 0.05$, compared with untreated cells. (**c**) Quantification results of recruitment of MSH2 and pol β to GAA repeats of the *FXN* gene in normal lymphoblasts. (**d**) Quantification results of recruitment of MSH2 and pol β to GAA repeats of the *FXN* gene in LoVo cells. Experiments were repeated for at least three times.

and expansion of one to two repeats (Fig. 5c, panel 2), whereas repair in the context of (CAG)$_{20}$ repeats led to a prominent deletion of two CAG repeats (Fig. 5d, panel 2). Consistent with our previous findings[48,53–55], deletions were the most prominent BER repair products, independent of the repeat sequence, when MSH2–MSH3 was absent from the reaction (Fig. 5c,d). Surprisingly, the addition of MSH2–MSH3 to the BER reactions largely attenuated deletions in the (GAA)$_{20}$ (Fig. 5c, panel 3) and (CAG)$_{20}$ substrates (Fig. 5d, panel 3). Rather, cooperation of the two proteins generated expansion of one to two repeats (Fig. 5d, panel 3), consistent with the size that is observed at premutation-length alleles in human patients[59]. PCR amplification of undamaged (GAA)$_{20}$ or (CAG)$_{20}$ substrates failed to show any repeat expansion or deletion products (Fig. 5c,d, panel 1), indicating that DNA instability depended on the lesion and BER. Thus, in the context of BER, MSH2–MSH3 suppressed deletion and promoted expansion via strand-displacement synthesis. Interestingly, some GAA repeat expansion occurred without MSH2–MSH3. However, CAG repeat expansion absolutely required MSH2–MSH3. This may result from more efficient synthesis of GAA than CAG repeats by pol β.

**MSH2–MSH3 stimulates pol β synthesis during BER.** We tested how MSH2–MSH3 acting together with pol β might suppress deletion and promote expansion. TNRs expand or contract if TNR loops form on the daughter strand or the template strand, respectively. Since deletion and expansion products were altered

by addition of MSH2–MSH3, we postulated that it might affect loop formation during pol β synthesis in a strand-specific manner. To test the role of MSH2–MSH3 in suppressing deletion, we used S1 nuclease to map a TNR loop that formed on the template strand of (GAA)$_{20}$ or (CAG)$_{20}$ substrates during BER (Fig. 6). The substrates were labelled at the 5′-end of the template strand (Fig. 6a,g), and the resulting single-strand cleavage patterns visualized the position and size of loops that formed at (GAA)$_{20}$ (Fig. 6a–f) or (CAG)$_{20}$ tracts (Fig. 6g–l). As before, pol β DNA synthesis was initiated by pre-cleaving the damaged strand using 10 nM APE1, and added were the purified components of the BER repair machinery without or with MSH2–MSH3 (Fig. 6). The schematic diagram of the results of each substrate product is below each figure (Fig. 6f,l).

In the absence of pol β and MSH2–MSH3, the APE1 nick site and small flap opened on breathing to expose around three repeats to the S1 nuclease, as indicated by S1 cleavage bands between +48 and +55 nt (Fig. 6b), and the template was eventually degraded. Addition of MSH2–MSH3 alone to the BER reaction had a strong protective effect (Fig. 6c), and blocked S1 cleavage at the APE1 incision site, preventing degradation of the template. MSH2–MSH3 appeared to cover the template and infrequently dissociate (Fig. 6c,f).

We next evaluated the S1 sensitivity of the template strand during active pol β synthesis. Surprisingly, in the absence of MSH2–MSH3, S1 cleavage generated a distinct ladder of bands between +53 and +23 below the APE1 incision product (Fig. 6d), indicating that the active pol β induced a large single-stranded region in the template strand (Fig. 6d). Indeed,

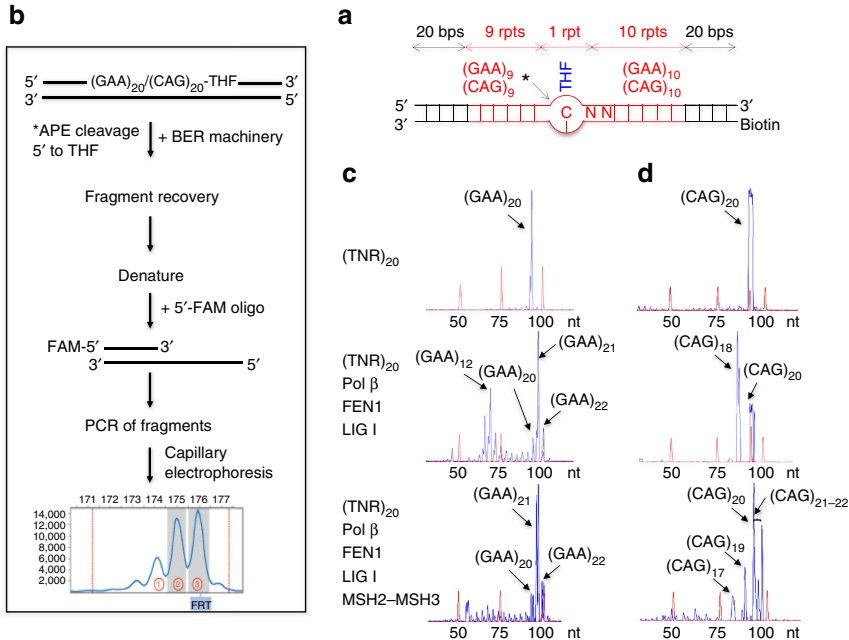

**Figure 5 | MSH2–MSH3 suppresses TNR deletion and promotes repeat expansion during BER.** The repair products were pulled down with avidin beads after the termination of the BER reaction. The repaired strands were subsequently separated from their biotinylated template strands by incubating with NaOH. The separated repaired strands were subsequently amplified using 5′-FAM-labelled primers and resolved by high-resolution capillary electrophoresis to define their length. (**a**) A schematic diagram of $(GAA)_{20}$ or $(CAG)_{20}$ substrate containing an abasic lesion. (**b**) A schematic diagram of the PCR amplification methodology and electrophoretic separation of the BER products. (**c,d**) GeneMapper scans of the ligated and amplified $(GAA)_{20}$ (**c**) or $(CAG)_{20}$ (**d**) products in the absence and presence of MSH2–MSH3. The top panel represents the size of the $(GAA)_{20}$ or $(CAG)_{20}$ templates without any damage or BER machinery added. The middle and lower panels illustrate DNA fragment analysis of the templates in **a** reconstituted with the BER machinery in the absence (middle) and presence (lower) of MSH2–MSH3. Key TNR sizes are indicated with black arrows. The red peaks in the scans indicate size standards, whose size is indicated below the scans. All experiments were performed in at least triplicate. rpt, repeat.

in reactions lacking deoxynucleoside 5′-triphosphates (dNTPs) and pol β synthesis, the S1 cleavage was suppressed (compare Supplementary Fig. 7b,c with Fig. 6d,f). Thus, pol β synthesis facilitated the formation of a large single-stranded loop or multiple small loops within the TTC tract (Fig. 6d,f). When MSH2–MSH3 and pol β were added together, however, the single-stranded region on the TTC template strand did not form, as judged by S1 cleavage (Fig. 6e,f). Thus, pol β could not efficiently copy the TTC repeats in the absence of MSH2–MSH3 and appeared to bypass them, forming a looped out region in the process (Fig. 6d,f). Similar results were obtained with the $(CAG)_{20}$ repeat substrate (Fig. 6g–l; Supplementary Fig. 7e,f). Consistent with this finding, active polymerase on the 1 nt gapped random sequence substrate, which does not form secondary structure, resulted in only a single product with 53 nt (Supplementary Fig. 8). Collectively, the results suggested that MSH2–MSH3 suppressed deletion by stimulating pol β to copy through the repeats on the template strand.

We tested whether MSH2–MSH3 could promote expansion by stimulating pol β DNA synthesis (Fig. 7). In this experiment, the damaged strand of $(GAA)_{20}$ and $(CAG)_{20}$ substrates was labelled at the respective 5′-ends, and their length was monitored during pol β DNA synthesis, without and with MSH2–MSH3 (Fig. 7a). APE1 cleavage produced the expected incision product (Fig. 7b, lane 2). Consistent with the S1 analysis, pol β was less processive in the absence of MSH2–MSH3 and extended only one or two TNRs from the APE1 cleavage site on both $(GAA)_{20}$ and $(CAG)_{20}$ substrates (Fig. 7b,c, lanes 3 and 8; Fig. 7d). Addition of MSH2–MSH3, however, stimulated pol β to copy the TNR tract and inserted up to seven GAA or six CAG repeats relative to pol β alone (Fig. 7b,c, compare lanes 4 and 9 with lanes 3 and 8; Fig. 7d). MSH2–MSH3 had a direct stimulatory effect on pol β, as

its DNA synthesis was not altered when MSH2–MSH3 was substituted by the same concentration of bovine serum albumin (BSA; Fig. 7b,c, lanes 5 and 10). The extra synthesis was not due to a contaminating DNA polymerase. In parallel reactions, we failed to observe any DNA primer extension on an open template from any of the components alone or when purified MSH2–MSH3 was added at concentrations up to 500 nM (Fig. 7f).

**Cooperation of MSH2–MSH3 and pol β promotes flap formation.** We have previously demonstrated that the damaged strand on a CAG template forms a displaced flap during pol β DNA synthesis, which provided a precursor for expansion[44]. To monitor the length of the flap during strand displacement, we labelled the 3′-end of the damaged strand (Fig. 7g), and measured the size of the flap during pol β DNA synthesis as judged by FEN1 cleavage. No flap cleavage was observed in the absence of FEN1 (Fig. 7h, lanes 1–3). When FEN1 was added in the absence of MSH2–MSH3, the most prominent flap lengths for the GAA template were between two and four repeats in the simple BER reactions (Fig. 7h, lane 4; Figure 7j), although flaps up to eight repeats formed infrequently. However, addition of MSH2–MSH3 significantly increased flap size such that the major lengths reached or exceeded eight repeats (Fig. 7h, lane 5; Fig. 7j). The increase in flap size depended on MSH2–MSH3, since its replacement with BSA abolished the effect (Fig. 7h, lane 6), and MSH2–MSH3 without or with pol β synthesis failed to cut the DNA (Fig. 7h, lane 7 and Supplementary Fig. 9, lanes 2 and 4).

Thus, MSH2–MSH3 both stimulated strand displacement and generated flaps that were significantly longer than the four to five repeats copied by pol β alone. Indeed, the difference between the

most prominent repeats synthesized by pol β (Fig. 7b, lane 4) and those removed by FEN1 (Fig. 7h, lane 5) predicted the one to three repeat expansions, which were present after ligation in the DNA fragment analysis (Fig. 5c, panel 3). The CAG templates showed similar properties. MSH2–MSH3 stimulated pol β to synthesize five to six extra repeats, while FEN1 removed only two repeats (Fig. 7i, lane 12; Fig. 7k), generating tracts that were longer than the original template. Collectively, MSH2–MSH3 not only prevented deletion by stimulating pol β synthesis of the TNRs on the template strand but also caused

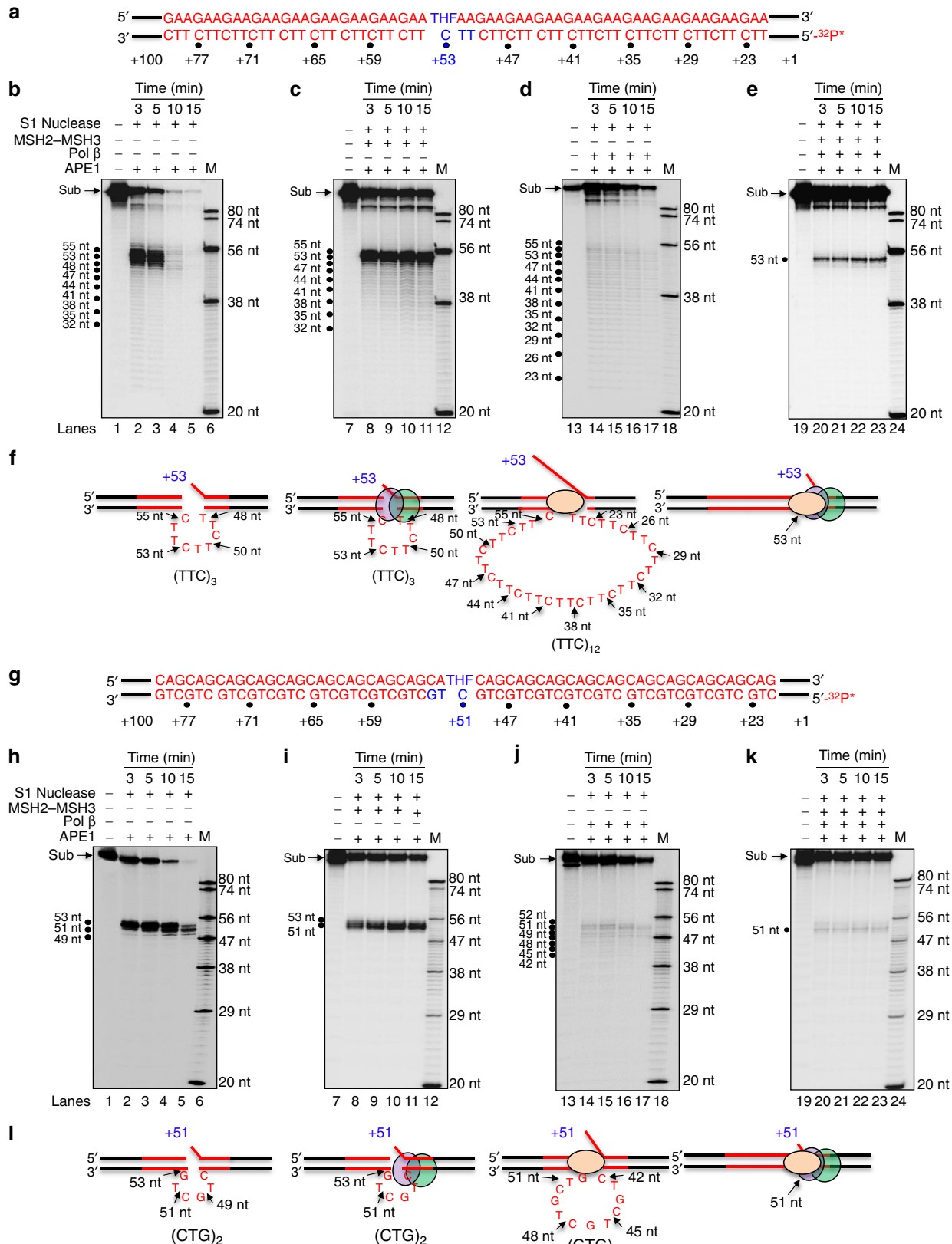

expansion by stimulating strand displacement of the damaged strand during BER.

## Discussion

*In vivo* and *in vitro* mechanistic testing of individual pathways has provided no consistent picture of how MMR causes expansion. We report here that MSH2–MSH3 acts as a key co-factor in the BER pathway, and in that context, promotes expansion of TNR tracts harbouring lesions that are canonical substrates for BER. Pol β makes a previously unrecognized physical complex with MSH2–MSH3, and without its help, pol β fails to complete DNA synthesis in the TNR loop and bypasses them on the template strand causing large deletions. MSH2–MSH3 enhances pol β loading onto TNRs, stimulates DNA synthesis through a TNR loop and enhances strand displacement. The results provide, to our knowledge, the first direct evidence for a hybrid mechanism that requires cooperation of both the MMR and the BER machinery to cause the expansion mutation, and explains the fact that both pathways are implicated *in vivo*. While *in vitro* BER alone supports primarily deletions, addition of MSH2–MSH3 switches the primary outcome to TNR expansion.

Taken together, the results support a 'toxic oxidation cycle' by a MMR–BER crosstalk mechanism[44] (Fig. 8, + MSH2–MSH3). After removal of the oxidative base, the residual 5′-flap becomes the loop precursor, whose incorporation by an 'in trans' endonuclease completes the expansion. The cycle can repeat with age. If MSH2–MSH3 is not available, pol β cannot traverse the repeats, which are then removed by 'in cis' endonuclease activity, resulting in deletion (Fig. 8, − MSH2–MSH3). We find size of the expansion is smaller than the size of the flap. Since it blocks FEN1 processing of Okazaki fragments[28], it is likely that MSH2–MSH3 will also block access to the TNR flap junction and will prevent its removal. In such a model, MSH2–MSH3 adjusts its position to optimally bind to the loop, and in the process, allows flap re-equilibration and partial re-binding (Fig. 8). A new and shorter flap is generated, with a junction that is suitable for FEN1 processing and ligation (alternate flap cleavage)[60] (Fig. 8). These steps are consistent with known features of MMR and BER. FEN1 cleavage requires access to a junction and to the 3′-end of the synthesized strand[61], MSH2–MSH3 alters the junction of a CAG hairpin[32], ligase requires FEN1 activity[60,62] and coordination of APE1, pol β and FEN1 modulates CAG repeat expansion[56,60,63].

The crosstalk pathway that we describe resolves at least some of the controversial issues relevant to the role of MMR in expansion. Plasmids harbouring preformed hairpins undergo small insertions and deletions in cell extracts lacking MSH2, implying that it is not needed for loop repair. However, we find that the pol β•MSH2–MSH3 complex plays a definitive role in

loop formation during BER, which will require MSH2–MSH3 if the loop is not preformed. It has been debated whether loop resolution occurs through a canonical MMR pathway, or through an alternative pathway. Our results suggest that both resolution mechanisms are likely to have roles[64]. At a CTG hairpin, MSH2–MSH3 recruitment of MLH1–PMS2 results in endonucleolytic incision both 'in cis' and 'in trans' to a small loop lesion[65]. 'In trans' clipping by MLH1–PMS2 or other MutL-like endonucleases resolves the hairpin, and the extruded loop provides a template for gap-filling synthesis, leading to expansion. However, this process becomes inefficient when TNR loop sizes are larger than four repeats, which are often generated in cells during excision repair. Indeed, in our hands, the flaps generated at 20 TNRs exceed 8 repeats (Fig. 7) and are likely to increase as the TNR tract lengthens.

Alternative mechanisms for resolution are as yet unknown. However, the ability of MSH2–MSH3 to influence the outcome of BER implies a broader role in directing choice among redundant pathways for removing oxidative DNA damage. For example, in mice, CSB *in vivo* protects CAG repeats from expansion, while OGG1 tends to expand them[40]. A mechanism for CSB/OGG1 antagonism is unclear[66]. Although CSB is part of the TCR machinery, it also interacts with BER machinery such as APE1 (ref. 67) and poly (ADP-ribose) polymerase-1 (ref. 68). Thus, CSB may co-opt MSH2–MSH3 and prevent its ability to stimulate pol β, favouring deletions via BER. Such crosstalk may explain the protective effects of CSB and observations that treatment with oxidizing agents can induce expansions and contractions[41,48]. This kind of pathway choice is also predicted to be sensitive to the tissue-specific level of the available repair machinery, as has been noted for BER[58]. We have yet to test whether MSH2–MSH6 acts as a scaffold for pol β, but it is also possible, given their structural similarity, that MSH2–MSH6 participates in crosstalk by modifying pol β-MSH2–MSH3-mediated TNR instability.

Whatever the detailed mechanism, our results provide evidence that cells can implement crosstalk strategies to cause expansion, and share machinery when lesion resolution is difficult. Pathway crosstalk provides latitude in correcting a lesion by whatever machinery a cell may have available or that best fits the situation. The importance of crosstalk pathways cannot be overestimated, as the efficiency of modulating the TNR tract length determines whether repair is error-free or error-prone, and whether the biological outcome is genetic integrity or fatal disease.

## Methods

**Materials.** DNA oligonucleotides were synthesized by Integrated DNA Technologies Inc. (Coralville, IA). The radionucleotide [γ-$^{32}$P] ATP (6,000 Ci mmol$^{-1}$) and cordycepin 5′-triphosphate 3′-[α-$^{32}$P] (5,000 Ci mmol$^{-1}$) were purchased from PerkinElmer Inc. (Boston, MA). Micro Bio-Spin 6 chromatography columns were from Bio-Rad Laboratories (Hercules, CA). dNTPs were from Roche Diagnostics (Indianapolis, IN). T4 polynucleotide kinase and terminal nucleotidyltransferase were from Fermentas (Glen Burnie, MD). DNase I

**Figure 6 | Pol β cannot traverse the TNRs in a TNR loop on the template strand during BER without MSH2–MSH3.** S1 nuclease analysis of (GAA)$_{20}$ or (CAG)$_{20}$ templates with time before or during pol β DNA synthesis was resolved by PAGE. (**a**) The (GAA)$_{20}$ substrate was $^{32}$P-labelled on the 5′-end of the template strand. The S1 products of **a** identify the unpaired nucleotides on the template strand. The time of digestion is indicated in minutes. (**b,c**) S1 nuclease analysis of the starting (GAA)$_{20}$ template (**a**) after APE1 cleavage without (**b**) or with MSH2–MSH3 (**c**). (**d,e**) The same as **b,c** except that pol β is added to the reaction to initiate the DNA synthesis reaction in the absence (**d**) or presence (**e**) of MSH2–MSH3. Lanes 1, 7, 13 and 19 are the undigested substrate. The machinery present in each reaction is indicated above the gels; ( + ) is the presence of the component and ( − ) is the absence of the component. The sites of S1 cleavage are indicated on the left of the gel with major cut sites indicated by black dots, and correspond to the numbering system in **a**. The sizes of the synthesized DNA markers (M) are indicated to the right of the gel. The concentrations of the reagents in the reaction were as follows: substrate (100 nM), APE1 (10 nM), MSH2–MSH3 (100 nM), pol β (10 nM) and S1 nuclease (12 U). In all reactions, the substrates and BER components were pre-incubated with 10 nM APE1 to generate the 5′-end for pol β extension before digestion with 12 U of S1 nuclease at the indicated time intervals at 37 °C. **f** is the schematic summary of the S1 nuclease digestion using the numbering system in **a**. The black arrows and numbers indicate the positions of S1 cleavage. The purple and green balls are the MSH2–MSH3 heterodimer; the orange ball is pol β. Red are the TNRs and black are the random sequences. (**g-l**) Same as in **a–f** for the (CAG)$_{20}$ substrate. Experiments were repeated in triplicate.

was purchased from New England Biolabs (Ipswich, MA). S1 nuclease was from Promega Corporation (Madison, WI). All other reagents were purchased from Sigma-Aldrich (St. Louis, MO) and Thermo Fisher Scientific (Pittsburgh, PA). Purified recombinant human apurinic/APE1, pol β, FEN1 and DNA LIG I were expressed and purified as described[63]. In brief, APE1, his-tagged pol β and FEN1 were expressed in *Escherichia coli* BL21(DE3) (Thermo Scientific, Rockford, IL),

whereas his-tagged LIG I was expressed in *E. coli* BL21(DE3) AI strain (Thermo Scientific). The expression of the proteins was induced by 0.5 mM isopropyl-1-thio-β-d-galactopyranoside (APE1) or 1 mM (pol β and FEN1) for 3.5 h at 37 °C. LIG I expression was induced with 1 mM isopropyl-1-thio-β-d-galactopyranoside along with 0.2% (w/v) ʟ-arabinose for 24 h at 18 °C. Cells were lysed with a French press cell disruptor (Glen Mills, Clifton, NJ) in lysis buffer containing 30 mM

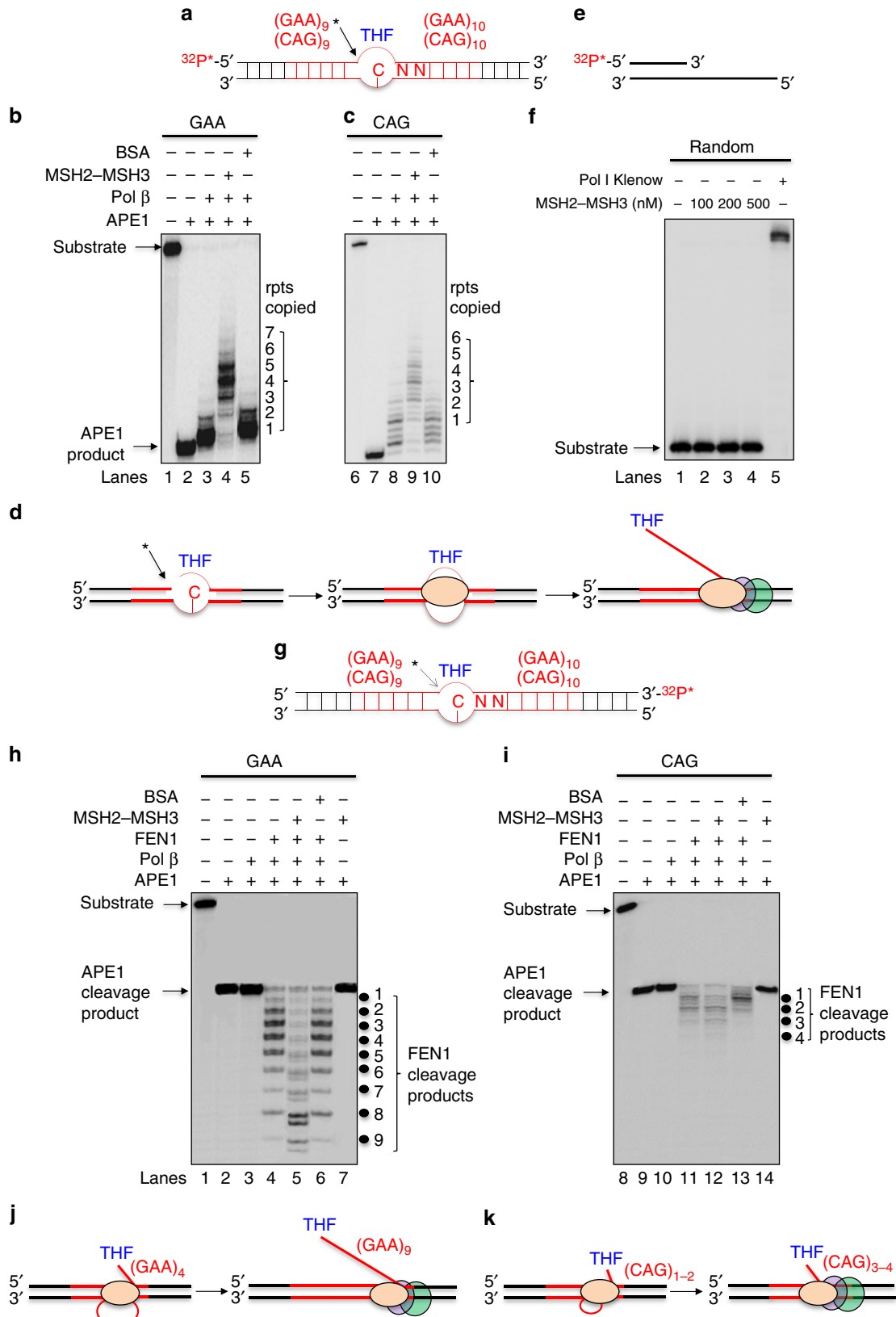

HEPES, pH 7.5, 30 mM KCl, 1 mM dithiothreitol (DTT), 1 mM EDTA, 1 mM phenylmethylsulfonyl fluoride and 0.5% inositol with proteinase inhibitors. The supernatant was recovered after centrifugation at 18,000$g$ at 4 °C and loaded sequentially onto a Sepharose Q column (GE Healthcare, Piscataway, NJ) or nickel-nitrilotriacetic acid (Ni-NTA) column (Qiagen, Valencia, CA; his-pol β), CM sepharose column (GE Healthcare), phenyl sepharose column (GE Healthcare) and Mono-S column (GE Healthcare) operated by an AKTA Fast Protein Liquid Chromatography system (GE Healthcare). For purification of his-LIG I, the supernatant was sequentially loaded onto a P11 phosphocellulose column (Whatman-GE Healthcare) and a Ni-NTA column. Purified proteins were aliquoted and frozen at − 80 °C until further use.

**Oligonucleotide substrates.** DNA oligonucleotide substrates containing a THF, an abasic site analogue, were designed to mimic an abasic site that occurs in a (GAA)$_{20}$ or (CAG)$_{20}$ repeat tract. The guanines in the tenth repeat unit of (GAA)$_{20}$ or (CAG)$_{20}$-containing substrates were substituted with a THF residue. Substrates were constructed by annealing an oligonucleotide with a THF residue to its template strand at a molar ratio of 1:2. Substrates mimicking the abasic site-containing intermediates preincised by APE1 were constructed by annealing the upstream strand and the downstream strand with a 5′-phosphorylated THF residue with the template strand at a molar ratio of 1:1:2. A DNA fragment that contained (GAA)$_{20}$ or (CAG)$_{20}$ without DNA lesions was used as a size marker for DNA fragment analysis. The sequences of the oligonucleotides are listed in Supplementary Table 1.

**MSH2–MSH3 purification.** Human MSH2 and His-tagged MSH3 were overexpressed in SF9-insect cells using a pFastBac dual-expression system (GIBCO-BRL), and purified as follows[11,30]. MSH2 and his-tagged MSH3 were overexpressed in SF9-insect cells using a pFastBac dual-expression system. Briefly, the supernatant was loaded onto a 5 ml HiTrap chelating column (GE Healthcare) charged with Ni-NTA affinity column and equilibrated with lysis buffer. The bound proteins were then eluted with a 25 ml 20–200 mM imidazole gradient. The peak fractions containing the MSH2/MSH3 (eluted at 140 mM imidazole) and were then loaded onto a Mono P and HiTrap Heparin column (GE Healthcare) connected in tandem and equilibrated in column buffer (25 mM HEPES NaOH, pH 8.1, 0.1 mM EDTA, 10% glycerol (v/v) and 1 mM DTT) containing 300 mM NaCl. The MSH2/MSH3 containing fractions were then applied to MonoQ (GE Healthcare). MSH2/MSH3 fractions were stored in 20% glycerol (v/v), aliquoted and frozen at − 80 °C.

**Gel mobility shift assay for detecting protein–DNA complexes.** Substrates (25 nM) were incubated with 25 nM APE1 for 15 min at 37 °C in reaction buffer with 5 mM Mg$^{2+}$ to generate 1-nt gap intermediates. Reactions were then incubated with 10 nM pol β in the absence and presence of 100 nM MSH2–MSH3 for 10 min on ice. Protein–DNA complexes were separated from the free DNA by 1% agarose–0.1% polyacrylamide gel electrophoresis (PAGE) at 4 °C (ref. 56).

**DNase I footprinting of binding sites of pol β and MSH2–MSH3.** DNase I footprinting was performed to identify the binding of pol β and MSH2–MSH3 to (GAA)$_{20}$ or (CAG)$_{20}$ substrates. Ten microlitres of binding reactions was assembled on ice with pol β (50, 100 nM), MSH2–MSH3 (100 and 200 nM) and 25 nM $^{32}$P-labelled DNA substrate in 10 mM HEPES, pH 8.1, 110 mM NaCl, 1 mM DTT, 0.1 mg ml$^{−1}$ BSA, 10% (v/v) glycerol and 0.25% (w/v) inositol with 5 mM Mg$^{2+}$, 2 mM ATP and 2 mM ADP. The reactions were made with pol β or MSH2–MSH3 added last. The binding reactions were incubated on ice for 15 min. Subsequently, the protein–DNA complexes were subjected to DNase I digestion in 20 μl reaction buffer that contained 10 mM Tris–HCl, pH 7.6, 2.5 mM MgCl$_2$ and 0.5 mM CaCl$_2$. The protein–DNA complexes were incubated with 0.008 U DNase I at 37 °C for 10 min. Enzyme digestion reaction was terminated by transferring to 95 °C for

10 min in 20 μl of stopping buffer containing 95% formamide and 2 mM EDTA. Substrates and nuclease digestion products were separated by 15% urea-denaturing PAGE and detected using a Pharos FX Plus PhosphorImager from Bio-Rad Laboratories. Synthesized DNA size markers were used to indicate the size of nuclease cleavage products.

***In vitro* reconstituted BER assay.** Substrates (25 nM) were pre-incubated with 25 nM APE1 at 37 °C for 15 min to generate single-strand DNA break intermediates before incubation with 100 nM MSH2–MSH3 on ice for 30 min. Subsequently, BER of a THF residue was reconstituted by incubating APE1 precut substrates with indicated concentrations of BER enzymes at 37 °C for 15 min in a 20-μl reaction mixture that contained reaction buffer, which was composed of 10 mM HEPES, pH 8.1, 110 mM NaCl, 1 mM DTT, 0.1 mg ml$^{−1}$ BSA, 10% (v/v) glycerol and 0.25% (w/v) inositol with 50 μM dNTPs, 5 mM Mg$^{2+}$, 2 mM ATP and 2 mM ADP. The reactions were terminated by transferring to 95 °C for 10 min. To isolate repair products, the template strand of the substrate was biotinylated at the 5′-end. Repair products were incubated with avidin agarose beads (Pierce-Thermo Scientific) in binding buffer that contained 0.1 M phosphate, 0.15 M NaCl, pH 7.2 and 1% (v/v) Nonidet P-40 at 4 °C for 2 h with rotation. Agarose beads were centrifuged at 2,000$g$ for 1 min and were washed three times with binding buffer. Repaired strands were then separated from their template strands by incubating with 0.15 M NaOH for 15 min with rotation under room temperature and centrifugation at 2,000$g$ for 2 min. Repaired strands were then precipitated with ethanol, dissolved in TE buffer and stored at − 20 °C for subsequent repeat sizing analysis.

**Sizing analysis of TNR length.** Repair products resulting from *in vitro* BER in the context of (GAA)$_{20}$ and (CAG)$_{20}$ repeats were amplified by PCR with a forward primer (5′-CGA GTC ATC TAG CAT CCG TA-3′) and a reverse primer tagged by a 6-carboxyfluorescein (5′-6-FAM-CA ATG AGT AAG TCT ACG TA-3′). PCR amplification was performed under the following conditions: 95 °C for 10 min, 1 cycle; 95 °C for 30 s, 50 °C for 30 s and 72 °C for 1.5 min, 35 cycles; 72 °C for 1 h. The 6-carboxyfluorescein-labelled PCR products were then subjected to capillary electrophoresis using an ABI 3130XL Genetic Analyzer (Applied Biosystems, Foster City, CA; Florida International University DNA Sequencing Core Facility). The size of repair products was determined by DNA fragment analysis using the GeneMapper version 5.0 software (Applied Biosystems). Size standards, MapMarker 1000 (Bioventures, Murfreesboro, TN), were run in parallel with PCR-amplified repair products.

**S1 nuclease digestion.** Formation of repeat hairpin or loop structures in the template strand was probed by incubating 12 U or 10 U S1 nuclease with 25 nM substrates that contained (GAA)$_{20}$ and (CAG)$_{20}$ repeats. Substrates containing a THF residue were pre-incubated with 10 nM APE1 in the absence or presence of 10 nM pol β at 37 °C for 30 min. The reactions with MSH2–MSH3 were performed by incubating APE1 precut substrates with 100 nM MSH2–MSH3 complex on ice for 30 min before incubation with S1 nuclease. The 10-μl reaction mixture was assembled in reaction buffer containing 50 mM sodium acetate (pH 4.5), 280 mM NaCl and 4.5 mM ZnSO$_4$. The reaction was incubated at 37 °C for 3, 5, 10 and 15 min, and subsequently subjected to protease K digestion at 55 °C for 30 min. Reaction mixtures were subjected to 95 °C for 10 min for denaturing DNA. Substrates and nuclease digestion products were separated by 15% urea-denaturing PAGE and detected by a PhosphorImager. Synthesized DNA size markers were used to indicate the size of nuclease cleavage products.

**Enzymatic activity assay.** Pol β DNA synthesis during BER was measured in the absence or presence of MSH2–MSH3 using 25 nM oligonucleotide substrates containing (GAA)$_{20}$ or (CAG)$_{20}$ with a THF residue as shown in Supplementary Table 1. The effects of MSH2–MSH3 on pol β DNA synthesis activity were

**Figure 7 | MSH2–MSH3 stimulates pol β DNA synthesis and enhances TNR flap size in the context of BER.** (**a**) The schematic representation of the (GAA)$_{20}$ or (CAG)$_{20}$ substrates with a $^{32}$P-label on the 5′-end of the damaged strand. (**b**) The labelled bands correspond to the length of pol β extension products in the absence or presence of MSH2–MSH3. The (GAA)$_{20}$ (**b**) or (CAG)$_{20}$ (**c**) substrates (25 nM) were incubated at 37 °C with 25 nM APE1 and 10 nM pol β in the absence or presence of 100 nM MSH2–MSH3. The machinery present in each extension reaction is indicated above the gels; ( + ) is the presence of the component and ( − ) is the absence of the component. BSA substitutes for MSH2–MSH3 in reactions in lanes 5 and 10. The number of repeats added is indicated to the right of the gels. (**d**) A schematic illustration of the results from **b** and **c**. (**e**) A schematic diagram of an open-template substrate containing random DNA sequence $^{32}$P-labelled at the 5′-end of the primer. (**f**) DNA synthesis activity from purified MSH2–MSH3 proteins was measured by incubating the open-template substrate shown in **e** without (lane 1) or with 100–500 nM of MSH2–MSH3 (lanes 2–4). Lane 5 represents reaction mixtures with 5 U Pol I Klenow fragment as a positive control. An extension product is observed only in lane 5. (**g–j**) The effect of MSH2–MSH3 on the size of the displaced strand during active BER after APE1 cleavage and pol β extension. (**g**) Schematic diagram of the (GAA)$_{20}$ or (CAG)$_{20}$ substrate illustrating the THF site and the position of $^{32}$P-labelling at the 3′-end of the damaged strand and the APE1 site (black arrow and star). Pol β DNA synthesis on the substrate generates displaced flaps, whose size is measured by FEN1 flap cleavage. (**h,i**) FEN1 cleavage products after pol β DNA synthesis of the (GAA)$_{20}$ (**h**) or (CAG)$_{20}$ (**i**) substrate. The concentration of FEN1 in the reaction is 10 nM, and all other reagents are the same as in **b** and **c**. Black dots and numbers to the right of the gel indicate the sizes of the FEN1 cleavage products. (**j,k**) Schematic representations of the results of **h** and **i**. Experiments were done in triplicate. rpts, repeats.

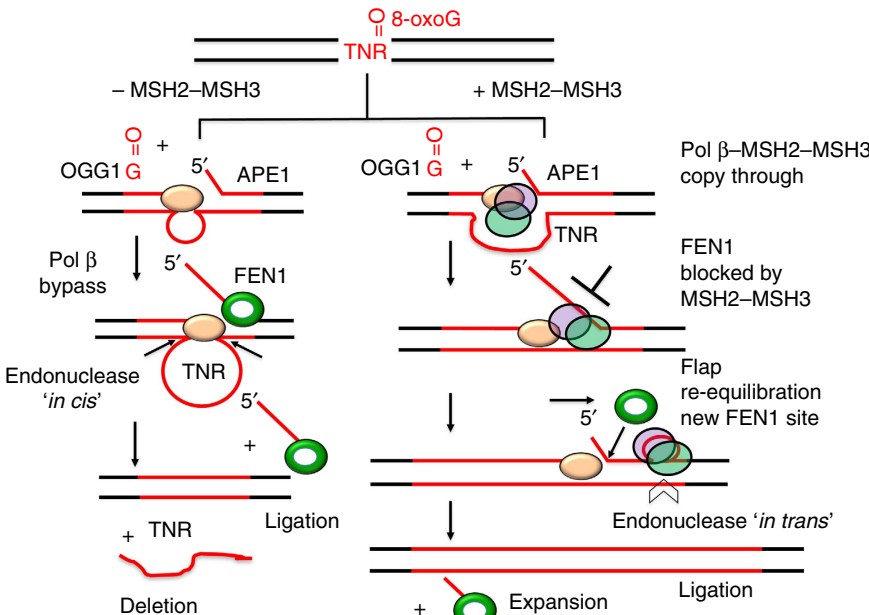

**Figure 8 | Model for a 'toxic oxidation' cycle by MMR–BER crosstalk.** MSH2–MSH3 promotes TNR expansion via suppression of repeat deletion during BER. Oxidative stress induces an oxidized DNA base in a TNR tract such as 8-oxoG. OGG1 removes the 8-oxoG and leaves an abasic site that is subsequently 5′-incised by APE1. The pol β•MSH2–MSH3 complex loads onto DNA at the APE1 incision site, and promotes DNA synthesis and flap formation. MSH2–MSH3 inhibits FEN1 removal of the flap. However, MSH2–MSH3 interaction with the loop reorients MSH2–MSH3 and allows flap re-alignment on the damaged strand to generate a new and shorter flap suitable for FEN1 cleavage and ligation. Incorporation of the loop by an endonuclease results in expansion. In the absence of MSH2–MSH3 ( − MSH2–MSH3), pol β opens the template to generate single-strand DNA loop structure, and deletion of the template strand occurs after endonuclease excision. Thus MSH2–MSH3 suppresses deletion and promotes expansion.

examined at 37 °C for 15 min in a 20-μl reaction mixture that contained reaction buffer with 50 μM dNTPs, 5 mM $Mg^{2+}$, 2 mM ATP and 2 mM ADP. FEN1 cleavage assay was also conducted in the absence or presence of MSH2–MSH3 under the same conditions. Reaction mixtures were subjected to 95 °C for 10 min in 20 μl of stopping buffer containing 95% formamide and 2 mM EDTA to denature DNA. Repair intermediates and products were separated by 15% urea-denaturing PAGE and detected by a PhosphorImager.

**Cell culture.** Human lymphoblast cell lines GM02152 (a normal individual with 15 GAA repeats), GM16207 (FRDA patient with 280/830 GAA repeats) and GM13511 (HD patient with 45/47 CAG repeats) were purchased from Coriell Institute for Medical Research (Camden, NJ) and cultured in RPMI 1640 medium with 15% fetal bovine serum (FBS), 2.05 mM L-glutamine and 1% antibiotics (penicillin and streptomycin). LoVo (MSH2 deficient) cells, from American Type Culture Collection (Manassas, VA), were grown in DMEM with 10% FBS, 2.05 mM L-glutamine and 1% antibiotics (penicillin and streptomycin). Cells were grown at 37 °C under 5% $CO_2$.

**Co-IP and immunoblotting.** A total of $8 \times 10^6$ lymphoblasts derived from a normal individual were treated with either 0.5 mM $K_2CrO_4$ or 10 mM $KBrO_3$ for 2 h, respectively. Untreated LoVo cells were used as a control. After treatment, cell lysates were prepared as described[54] Briefly, cells were lysed in buffer containing 10 mM Tris–HCl, pH 7.8, 200 mM KCl, 1 mM EDTA, 20% glycerol, 0.1% Nonidet P-40, 1 mM DTT and protease inhibitors with rotation at 4 °C for 2 h. Lysed cells were then subjected to centrifugation at 18,000g for 30 min. The supernatant was collected as cell lysates.

Subsequently, cell lysates were incubated with 100 μl protein A plus agarose (Pierce-Thermo Scientific) at 4 °C for 2 h with rotation. Cell lysates were further incubated without or with 1 μg of rabbit anti-pol β antibodies (ab26343, Abcam, Cambridge, MA), 1 μg of rabbit anti-MSH2 antibodies (ab16833, Abcam) or 1 μg of rabbit IgG (ab37451, Abcam) at 4 °C overnight with rotation, respectively. Cell lysates were then incubated with 50 μl protein A plus agarose at 4 °C for an additional 2 h with rotation, followed by three washes at 4 °C in washing buffer that contained 20 mM HEPES, pH 7.5, 150 mM NaCl, 1% NP-40 and 2 mM EDTA. Bound proteins were eluted by heating in SDS-loading buffer at 50 °C for 10 min and were subsequently subjected to SDS–PAGE and immunoblotting with rabbit anti-human pol β antibodies (1:1,000; ab175197, Abcam), mouse anti-human MSH2 antibodies (1:500; ab52266, Abcam) or rabbit anti-human MSH3 antibodies (1:800; ab154486, Abcam), followed by incubation with goat anti-rabbit IgG (1:10,000; ab6721, Abcam) or rabbit anti-mouse IgG (1:7,000; ab6728, Abcam) and chemiluminescent analysis

(Pierce-Thermo Scientific). To further confirm the direct interaction between pol β and MSH2, co-IP was also conducted with purified enzymes. A concentration of 100 nM purified pol β was incubated with 300 nM purified MSH2–MSH3 on ice for 30 min in 20-μl reaction mixture that contained reaction buffer with 50 μM dNTPs, 5 mM $Mg^{2+}$, 2 mM ATP and 2 mM ADP. Reaction mixtures were further incubated without or with 1 μg of rabbit anti-pol β antibodies (ab26343, Abcam), or 1 μg of rabbit anti-MSH2 antibodies (ab16833, Abcam) or 1 μg of rabbit IgG (ab37451, Abcam) at 4 °C for 2 h with rotation, respectively. Reaction mixtures were then incubated with 20 μl protein A plus agarose at 4 °C for an additional 2 h with rotation, followed by three washes at 4 °C in washing buffer that contained 20 mM HEPES, pH 7.5, 150 mM NaCl, 1% NP-40 and 2 mM EDTA. Bound proteins were eluted by heating in SDS-loading buffer at 50 °C for 10 min and were subsequently subjected to SDS–PAGE and immunoblotting with rabbit anti-human pol β antibodies (1:1,000; ab175197, Abcam) or mouse anti-human MSH2 antibodies (1:500; ab52266, Abcam), followed by incubation with goat anti-rabbit IgG (1:10,000; ab6721, Abcam) or rabbit anti-mouse IgG (1:7,000; ab6728, Abcam). Proteins were then detected by chemiluminescence.

**Immunocytochemistry and pixel analysis.** The protein–protein interaction between pol β and MSH2–MSH3 in lymphoblasts was determined by immunocy-tochemistry. Normal lymphoblasts (GM02152) and FRDA patient lymphoblasts (GM16207) were treated with 0.3 mM $K_2CrO_4$ or 30 mM $KBrO_3$ for 2 h. Untreated cells served as a negative control. After treatment, cells were pelleted by cen-trifugation at 1,000g for 3 min at room temperature. Cell pellets were washed with PBS, resuspended and fixed in three volumes of 4% paraformaldehyde (in PBS), and incubated at room temperature for 10 min. Fixed cells were pelleted by centrifuga-tion at 1,000g for 3 min at room temperature, and cell pellets were washed with PBS. Cells were then permeabilized by resuspension in three volumes of PBS containing 0.1% Triton X-100 and incubated at room temperature for an additional 10 min and were pelleted by centrifugation at 1,000g for 3 min at room temperature. Cell pellets were washed with PBS before incubation with antibodies for immunofluorescence.

Primary antibodies were as follows: rabbit anti-MSH2 polyclonal antibody (1:200, (N20), sc494, Santa Cruz Biotechnology, Dallas, TX) and mouse monoclonal anti-DNA polymerase β antibody (1:200; #ab1831, Abcam). Secondary antibodies were as follows: donkey anti-rabbit 568 (donkey anti-rabbit IgG (H + L) secondary antibody, Alexa Fluor 568 conjugate 1:250, #A10042, Life Technologies, Carlsbad, CA) and goat anti-mouse 488 (goat anti-mouse IgG (H + L) secondary antibody, Alexa Fluor 488 conjugate 1:250, #A11001, Life Technologies, Carlsbad, CA. Antibody dilutions were in PBS containing 3% BSA. Antibody incubations were for 1 h at room temperature. 4,6-Diamidino-2-phenylindole stain (Life Technologies, Carlsbad, CA) was added to a final concentration of 0.125 ng μl$^{-1}$ to

visualize the nuclei. Cells were mounted onto glass slides with ProLong Gold Antifade mounting media (Molecular Probes, Eugene, OR). Slides were imaged on a Zeiss 710 confocal system equipped with an inverted Zeiss Observer microscope and a LD-C Apochromat $\times 100$ 1.1 numerical aperture oil-immersion objective (Thornwood, NY). Pixel analysis was performed with Zeiss Zen software using the following criteria. Optical sectioning at 1.0 µm, and a pinhole set to provide 1 a.u. (0.7 µm sections using Plan apochromat $\times 100/1.4$ oil differential interference contrast objective). The lasers: argon (458, 488 and 514 nm), DPSS 5610-10 (561 nm), HeNe (633 nm), Diode 405-1 (405 nm); the filters: MBS 458/514 (visible light), MBS 405 (invisible wavelengths). The pixel size (on the sample): $514 \times 514$ pixels per frame.

**Chromatin immunoprecipitation.** A total of $8 \times 10^6$ lymphoblasts from a normal individual and a HD patient were treated with either 0.5 mM $K_2CrO_4$ or 10 mM $KBrO_3$ for 2 h. Untreated cells were employed as a negative control. After treatment, cells were washed twice with PBS and pelleted by centrifugation at 1,000$g$ for 3 min at room temperature. Cell pellets were then resuspended in complete culture medium that contained RPMI 1640 medium with 15% FBS. Formaldehyde (36.5%) was added to culture medium to a final concentration of 1% for the crosslinking reaction. Cells were incubated with formaldehyde at 37 °C for 30 min. Crosslinking was quenched by the addition of 1 M glycine to a final concentration of 125 mM and incubation with shaking for 5 min at room temperature. Cells were then pelleted by centrifugation at 500$g$ for 4 min at 4 °C and washed twice by ice-cold $1 \times$ PBS containing protease inhibitors (Roche Diagnostics). Washed cell pellets were further resuspended in lysis buffer that contained 1% (w/v) SDS, 10 mM EDTA and 50 mM Tris–HCl, pH 8.0, with protease inhibitors. Cell suspensions were incubated on ice for 10 min to release crosslinked chromatins before sonication. Cell lysates were further subjected to sonication for 15 cycles of 30 s ON and 30 s OFF for each cycle at 4 °C with Bioruptor ultrasonicator (Diagenode, Denville, NJ). The supernatant of the sheared cell lysate was separated from cell debris by centrifugation at 18,000$g$ for 10 min at 4 °C. The supernatant was further diluted 10-fold with ice-cold ChIP dilution buffer that contained 1% (v/v) Triton X-100, 1.2 mM EDTA, 167 mM NaCl and 16.7 mM Tris–HCl, pH 8.0, with protease inhibitors. 150 µl of the diluted supernatant was set aside as INPUT for total DNA control. The remaining lysates were initially incubated with sheared salmon sperm DNA-coated protein A sepharose (Life Technologies, Grand Island, NY) for 2 h at 4 °C with rotation. Subsequently, the lysates were divided into equal aliquots as the No-Ab control and immunoprecipitates (IPs) with pol β and MSH2, respectively. For each IP, the diluted chromatin solutions were incubated overnight with 2 µg rabbit anti-human pol β antibody (kindly provided by Dr. Samuel H. Wilson at the National Institute of Environmental Health Sciences, National Institutes of Health, Research Triangle Park, NC) or 2 µg rabbit anti-human MSH2 antibodies (ab16833, Abcam) at 4 °C with rotation. Subsequently, IP for pol β, MSH2 and the No-Ab control were incubated with sheared salmon sperm DNA-coated protein A sepharose beads for 2 h at 4 °C with rotation. IP bound to protein A sepharose beads was pelleted with centrifugation at 500$g$ for 2 min and washed two times with low-salt washing buffer that contained 150 mM NaCl, 0.1% (w/v) SDS, 1% (v/v) Triton X-100, 2 mM EDTA and 20 mM Tris–HCl, pH 8.0, followed by high-salt washing buffer that contained 500 mM NaCl, 0.1% (w/v) SDS, 1% (v/v) Triton X-100, 2 mM EDTA and 20 mM Tris–HCl, pH 8.0, and finally washed by TE buffer. IP was then eluted from agarose beads by incubation with freshly prepared elution buffer that contained 1% (w/v) SDS and 0.1 M NaHCO$_3$. Subsequently, IP was subjected to crosslinking reversal with 0.2 M NaCl and subsequent incubation at 65 °C for 6 h. Released DNA was cleaned up with proteinase K digestion at 45 °C for 2 h and phenol/chloroform extraction. DNA was then recovered by ethanol precipitation. The precipitated DNA were dissolved in TE buffer and used for subsequent quantitative PCR.

**Quantitative real-time PCR and data analysis.** Quantitative PCR was performed by using SYBR Green Supermix (Bio-Rad Laboratories) in a 20-µl reaction according to the manufacturer's protocols. Samples were amplified using a CFX Connect Real-Time PCR Detection System from Bio-Rad Laboratories. GAA repeats in *FXN* gene were amplified by a forward primer (5′-GGG ATT GGT TGC CAG TGC TTA AAA G-3′) and a reverse primer (5′-CCT ATT TTT CCA GAG ATG CTG GGA AAT CC-3′). The amplification was carried out by the following PCR procedure: 98 °C for 2 min (initial denaturation), 98 °C for 20 s (denaturation), 65 °C for 3 min (annealing and extension), 40 cycles. The length of PCR products should be $(422 + 3n)$ bp ($n =$ number of GAA triplets). CAG repeats in *HTT* gene were amplified by a forward primer (5′-GCT CAG GTT CTG CTT TTA CCT GC-3′) and a reverse primer (5′-TGC AGG GTT ACC GCC ATC-3′). The amplification was carried out by the following PCR procedure: 98 °C for 2 min (initial denaturation), 98 °C for 20 s (denaturation), 51 °C for 1 min (annealing), 72 °C for 2 min, 40 cycles. The length of PCR products should be $(394 + 3n)$ bp ($n =$ number of CAG triplets). Ct values that were recorded in CFX Manage Software (Bio-Rad Laboratories) during PCR were used for performing quantification of data to evaluate the fold difference between experimental samples and normalized input. ΔCt [normalized ChIP] (normalized to the input samples) = Ct [ChIP] − (Ct [input] − Log$_2$ (input dilution factor)), where input dilution factor = (fraction of the input chromatin saved)$^{-1}$. In our experiments, the fraction of input chromatin used for further analysis was 150 µl, whereas the fraction used for each IP was 600 µl. The fraction for IP was four times of the input fraction. Thus, the input dilution factor was 4, and the equation was derived as:

ΔCt [normalized ChIP] = Ct [ChIP] − (Ct [Input] − Log$_2$ (4)). The input % for each sample was calculated as: input % = $2^{-ΔCt\,[normalized\,ChIP]} \times 100$. The 'input %' value represents the enrichment of pol β and MSH2 on the GAA repeats of *FXN* gene or the CAG repeats of *HTT* gene. Statistical analysis was performed using the GraphPad Prism 6 (Graphpad software, San Diego, CA). Statistical significant differences in the data were tested by standard two-way analysis of variance with Tukey's multiple comparison post tests. A significant difference was designated at $P < 0.05$.

**Data availability.** The authors declare that all data supporting the findings of this study are available within the article and its Supplementary Information files are available on request from the corresponding authors.

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

## Acknowledgements

We thank Samuel H. Wilson at National Institute of Environmental Health Sciences, National Institutes of Health for generously providing expression vectors for BER enzymes and antibodies against human pol β. This work was supported by National Institutes of Health grants ES023569 (to Y.Liu), ES020766 (to C.T.M.), NS060115 (to C.T.M.) and CA092584 (to C.T.M.).

## Author contributions

Y.Lai and Y.Liu conceived, performed the experiments and wrote the manuscript. C.T.M. supervised MSH2–MSH3 purification and pixel analysis, and contributed to writing. H.B. performed the immunocytochemistry and pixel analysis. N.L.S.C. purified MSH2–MSH3, and performed chromatography separation of the OGG1/MSH2–MSH3 complex. J.M.B. performed the experiments and contributed to writing. Z.Z. contributed to scientific discussions and study design.

## Additional information

**Competing financial interests:** The authors declare no competing financial interests.

