## [Peer review file · Nature Communications]

Reviewers' comments:

Reviewer #1 (Remarks to the Author):

In this manuscript, the authors aimed to characterize the contribution of msh2/msh3 proteins to the dynamics of trinucleotide repeats (TNR) expansion/deletion in the context of DNA polymerase beta-dependent BER. To this aim, the authors use an in vitro model based on a dsDNA template with a single synthetic abasic site flanked by several TNR (they also use two different triplet sequences). They showed that Pol beta and msh2 physically interacts both in vitro and in cell extracts and colocalize in the cell upon oxidative stress. By reconstituting the BER reaction in vitro with purified components, the authors show that this interaction modulates the ability of Pol beta to traverse the TNR, promoting limited expansion and suppressing deletion.

This is an important study, providing mechanistic insights into the established role of MSH2/MSH3 in TNR instability and suggesting a molecular cross-talk between BER and MMR.

To test their hypotheses, the authors used several complementary techniques, which were able to provide independent evidence for supporting their conclusions.

Overall, the authors' data seem robust and their conclusions logical. The proposed model provides several experimentally testable hypotheses, that may help, in future work, to further define this novel hybrid repair pathway.

There are a number of issues that should be addressed, in order to improve the clarity and robustness of the manuscript.

Major Points

1) Page 7, line 4. Figure 1. Why combination of APE1 with MSHs leads to a strong APE1-DNA complex (Fig. 1 b, lanes 4 and 10, lower band)? The authors do not comment, but this may be relevant. Would it be possible to hypothesize that MSH2/3 complex recognizes the abasic site, acting in concert with APE1? Maybe this stimulatory effect should be tested also in the context of APE1 incision of the abasic site.

2) Page 9, line 14 and Page 10, line 4. Fig. 3. It should be more clearly specified that fig. 3 e is the demonstration of the direct interaction between purified proteins, ruling out the presence of a bridging factor in the pulldowns with the extracts. However MSH3 was not tested here. Indeed, the fact that in LoVo no complex is detected proves that MSH3 alone is not sufficient and MSH2 is essential, but not that MSH2 and MSH3 are both present in the complex. Was MSH3 also pulled down in cells? The bandshift exps (Fig. 1) have been performed with both proteins. Is the bandshift reproducible then with MSH2 alone or is MSH3 in any case required?

MSH2/MSH6 complex may also act on this substrate and even microsatellite instability has been ascribed to MSH2/MSH3 complex, it remains an interesting question to be addressed whether MSH2 is a scaffold protein for Pol beta, allowing formation of alternative complexes with either MSH3 or MSH6. It would be interesting to test this, but at least the authors should discuss this possibility, also

in the context of the recent paper by Nakatani et al. *SciRep.* 2015, 5:11020.

3) Page 10, line 10. Along the line of the previous point, while the experiments in LoVo cells are pretty convincing, it should be better to have also a gain-of-function experiment, where LoVo are complemented with exogenous MSH2 to see whether colocalization is restored.

4) Page 10, line 14, Supp. Fig. 5. Actually treatment with K₂CrO₄ apparently caused a significant increase, higher than with KBrO₃ (longer tail in the pixel analysis panel), which was not apparent in fig. 3f. Is there a difference in the amount/kind of damages caused by these two agents?

5) Page 10, line 16-17. Here the authors make a general consideration on the possible role of these proteins in normal BER, specifically arguing that lack of colocalization of MSH2 and Pol beta reflects unrepaired oxidized bases. However, this was not proven. It is also unclear why the basal level of oxidative stress should be intrinsically higher in FRDA cells. Moreover, if the hypothesis of saturation of repair systems is true, an higher increase in 8oxoG accumulation should be observed in treated FRDA cells with respect to normal cells. This can be directly assessed by IF or 8oxoG quantification by ELISA. Lastly, it may be possible to speculate that saturation of repair systems may lead to accumulation of DNA damage, eventually converted into ssDNA breaks by incomplete repair. A neutral comet assay may also show whether there is an higher accumulation of SSBs in FRDA cells vs wild type, either untreated or after treatment.

6) Page 11, line 5. In Supp. fig. 6a, lane 4 pol beta seems associated to HTT CAG even in untreated conditions (if I am not mistaken of course). This is consistent with previous studies (e.g. *PLoS Genet.* 2009 Dec; 5(12): e1000749.), but it is not the same profile as shown in fig. 4a, lane 4. Maybe the authors could comment.

7) Page 13, line 2 - 4, figure 6. It appears that the loop is stabilized and the adjacent sequences protected by S1 digestion (compare 6b lanes 4,5 to 6c lanes 10, 11. This is relevant and should be discussed. Moreover, it is difficult to judge the effects of Pol beta alone, since very little digestion is seen in panel 6d. Comparing panels c and d clearly show the reduction of the prominent cleaved products at 53-55 (consistent with the lack of the short loop), but it is not clear why the authors postulate the appearance of a much larger loop, since only a slight increase in shorter fragments (23-2) is visible, but they were present also in panel b. This should be more clearly proven, maybe by repeating the S1 experiments and titrating different amounts of MSH2/MSH3 in order to see a more clear accumulation of the short products.

8) Page 13, line 13. The template strand downstream the C at position +1 is composed of a tract containing 10 CTT repeats. The products resolved on the gel show that Pol beta was able to perform strand displacement (or traverse the ss loop formed according to the hypothesis in fig. 6) copying either 1-2 or 6-7 of these repeats. There is no evidence that these will be added as extra repeats. So the word "insertion" or "added" as in the panels should be replaced by "copied". The authors may want to refer already here to the results shown in Fig. 7, with the Fen1 digestion, in support to their conclusions.

9) Page 13, lines 7 - 9. Here it is written that Pol beta was able to fill the gap (lines 7-8) in the

absence or presence of Pol beta (line 9)? Supplementary Figure 7 actually show a S1 digestion experiment where no major loop formation can be seen. However it is not immediately clear how gap filling activity can be judged from this experiment. The authors may want to perform the same experiment as shown in Fig. 7 of the paper (DNA synthesis) also on this random substrate to show precise gap filling by Pol beta.

10) Page 14, line 10. "no flaps were observed". It is not clear what the authors meant. Flaps are detected by Fen1 cleavage. In the absence of Fen1 no cleavage was observed, which does not mean there were no flaps. Maybe I am missing something here, and if so I apologize, but if the authors suppose that Fen1 itself promotes Flap formation, they should prove in some independent way that no flaps were present when Pol beta was included.

Also Page 14, line 16 "MSH2/MSH3 alone was ineffective at generating a flap". This is contradictory to the previous results of the S1 digestion. MSH2-3 likely generate a long flap, but cannot cleave it in the absence of Fen1. Maybe this can be rectified by writing "...alone cannot generate cleavage products..."

Minor points

1) Page 5, line 16. "...for the first time...". It is always better to avoid too firm statements. Adding "to the best of our knowledge" is suggested.

2) In the Introduction or Discussion, the authors may want to cite also Nakatani et al. SciRep. 2015, 5:11020 where msh2/msh3 knock down suppresses repeat expansion in cells (see also major point 2 for a possible specular role of MSH6).

3) In many places, the authors state that they pre-treated the substrate with APE1 to make the incision. However, such incision should be formally proven by running the substrate on a denaturing gel in order to see the accumulation of the 5'end labelled strand cleaved product. This is shown only in figure 7 b. At least the authors should specifically refer to such figure as a demonstration that APE1 can indeed cleave the synthetic abasic site.

4) Page 12, line 25. panels 6d and 6f represent schematic diagrams of the substrates. Maybe the authors meant to refer to the fragment analysis shown in figure 5? Then the authors should say so.

5) 80 references are probably too many for a research paper. I would suggest to reduce them to less than 70.

Reviewer #2 (Remarks to the Author):

This manuscript describes an interesting and mostly well-executed set of experiments that demonstrate that MutS β , a component of the mismatch repair and Pol β , a component the base excision repair pathways co-localize in vivo in response to oxidative damage and interact in vitro in such a way as to generate products that could result in expansions of GAA and CAG repeats in vivo. Their interaction in vitro increases the processivity of the normally not very processive Pol β enzyme. The authors' data suggest that in vivo this could reduce the generation of products that result in deletions and increase the generation of products that could in principle lead to expansions. This data provides a plausible explanation of the key steps of the mutational mechanism that gives rise to the Repeat Expansion Diseases.

Most of my concerns are relatively minor involving either confusing/incomplete descriptions or the use of overly broad statements that are not fully supported by the data. However, some data is unconvincing or open to other interpretation. For example, MSH2 and Pol β are said to co-localize in data shown Fig. 3f. However, the data provided show very little clear evidence of that, with both proteins showing a very wide distribution.

In addition, on Pg. 13 lines 4-12, it is claimed that the behavior of Pol β on a 1nt-gapped substrate containing a repeat is "remarkable" in that "Pol β was unable to traverse the repeat loop on the template" generating a large single stranded region that was not seen on a template that did not contain the repeat (Fig. 6 and Supplementary Fig. 7). This result is very difficult to understand in terms of the authors' explanation since it is unclear how the long regions of the repeat would become unpaired during the reaction given that Pol β is not very processive and the reaction contains no accessory proteins that may facilitate this unpairing. An alternative explanation is that with a poorly processive polymerase like Pol β , strand dissociation there is a lot of strand dissociation. However, only on a repeat-containing template is there efficient strand-slippage and repriming from the slipped position. Could it be that the large region of S1 sensitivity seen on the repeat template does not reflect a single large loop out as the authors suggest, but a series of small loop outs resulting from strand slippage and repriming at multiple places in the repeat tract. When the processivity of Pol β is increased in the presence of MutS β , then strand slippage is less likely to occur (while the likelihood of strand-displacement increases). Thus, it may be that the data in Fig. 6 reflects the possibility of strand slippage on a template containing repeats rather than Pol β being unable to traverse a loop generated on the template, a perhaps subtle but important difference that would not be so much remarkable, but consistent with expectations.

The remaining points are addressed more or less in the order in which they appear in the manuscript.

- 1) The title of the manuscript should be amended to include the words "in vitro" since no data are presented proving that the mechanism delineated in vitro is in fact what is responsible for expansion in vivo.
- 2) Pg. 6 (and the abstract). The phrase beginning with "Since genetic interactions of MSH2-MSH3, OGG1, NEIL1 and Pol β cause expansions in mice" should be rephrased. Each of those proteins have been shown to promote expansions however, strictly speaking no interactions genetic or otherwise

have until now been demonstrated.

3) Pg. 6. It is stated that there was little evidence for a direct interaction of MSH2-MSH3 with OGG1 on templates with an oxidized base. However, the data for this (Supplementary Fig. 2) does seem to show some evidence of interaction. Unfortunately since the experiment used a different methodology from that used to study the interaction MSH2-MSH3 with Pol β , a direct comparison is difficult.

4) Why does the CAG substrate binds less well to MutS β than the GAA template in Fig. 1b but not Fig. 1c? Also, it is not mentioned in the text but it is apparent from Fig. 1b, that MutS β also improves APE1 binding (again more to the GAA template than the CAG template). The nature of this binding should be addressed since it does not seem to involve an interaction of MutS β with APE1 that is stable enough to survive electrophoresis.

5) Could the authors comment on the fact that while it is true that the protection of the substrate by Pol β and MutS β is more dramatic on one side of the nick, there is a subtle but distinct change in the DNase I footprint seen downstream of the repeat, at least in the case of the GAA repeat (Supplementary Fig. 3)?

6) The authors should cite evidence for the statement that oxidative damage is elevated in FRDA patient cells to support their contention that the absence of increased colocalization of Pol β and MutS β in response to oxidative damage in patient cells reflects the elevated background levels of oxidative damage in these cells. They should also provide evidence that this same effect is seen in lines established from other FRDA patients.

7) Pg 11, line 16. A little more detail at this point as to how the experiment was done would make this paragraph easier to follow.

8) Pg. 12, lines 24-26. This section is difficult to follow and may contain errors.

9) Pg. 13, line 1-2. It is stated that addition of MutS β has little effect on the S1 digestion of the template in the absence of Pol β , however it seems clear that MutS β is protecting the region from further S1 digestion.

10) Pg. 13, line 6 (as well as the abstract (Pg. 2, line 6)). The phrase "consistent with a deletion" is too imprecise. It is more accurate to say that this situation could ultimately result in the production of a deletion.

11) Pg. 13, line 15. Strictly speaking the experiments in this section do not confirm the results of the previous experiments. Rather they are an evaluation of what is happening on the other strand.

12) Pg. 13, line 18 (as well as the abstract (Pg. 2, line 6) and Pg 15, line 9. It is not correct to say that Pol β failed to complete gap-filling synthesis in the absence of MutS β , but rather that synthesis was less processive.

13) Pg. 15, line 18. The details of the "toxic oxidation cycle" model are not well described. Additionally, since the term "toxic oxidation cycle" is sometimes used to describe a somewhat different model for the role of oxidative damage in repeat expansion, it might be better to come up with a different name for the authors' model.

14) Figure 8. The arrow indicating the site of the "trans" acting endonuclease in the model should be placed further 5' on the bottom strand shown in the right hand panel so that repair of the nick would entail replication of the loop introduced on the top strand during flap displacement synthesis.

Author response to reviewer comments

Reviewer #1 (Remarks to the Author):

This is an important study, providing mechanistic insights into the established role of MSH2/MSH3 in TNR instability and suggesting a molecular cross-talk between BER and MMR. To test their hypotheses, the authors used several complementary techniques, which were able to provide independent evidence for supporting their conclusions. Overall, the authors' data seem robust and their conclusions logical. The proposed model provides several experimentally testable hypotheses, that may help, in future work, to further define this novel hybrid repair pathway. There are a number of issues that should be addressed, in order to improve the clarity and robustness of the manuscript.

Major Points

1) *“Page 7, line 4. Figure 1. Why combination of APE1 with MSHs leads to a strong APE1-DNA complex (Fig. 1 b, lanes 4 and 10, lower band)? The authors do not comment, but this may be relevant. Would it be possible to hypothesize that MSH2/3 complex recognizes the abasic site, acting in concert with APE1? Maybe this stimulatory effect should be tested also in the context of APE1 incision of the abasic site.”*

We appreciate the reviewer's careful reading and helpful comments. It is unlikely that the lower band was APE1•MSH2-MSH3•DNA complex as it was also detected in the lanes without APE1 (Fig. 1c, lanes 3 and 8). Its presence in all lanes indicates that it is likely to be a small truncation of the oligonucleotide, which can be bound by MSH2-MSH3. We have noted the band in the Figure legend of Fig. 1 in the revised manuscript (Page 34, lines 20-22).

2) a) *“Page 9, line 14 and Page 10, line 4. Fig. 3. It should be more clearly specified that fig. 3e is the demonstration of the direct interaction between purified proteins, ruling out the presence of a bridging factor in the pulldowns with the extracts.”*

We appreciate the reviewer pointing this out. We agree with the reviewer that it was not clearly stated. In the revised manuscript, we have re-written the text according to the reviewers' suggestion on Page 9, lines 17-19 (red). It now reads “The interaction was observed in reactions containing the purified proteins, ruling out the possibility that a bridging protein mediates the formation of the MSH2-MSH3•pol β protein complex in the cell extracts.”

b) *“However MSH3 was not tested here. Indeed, the fact that in LoVo no complex is detected proves that MSH3 alone is not sufficient and MSH2 is essential, but not that MSH2 and MSH3 are both present in the complex. Was MSH3 also pulled down in cells?”*

Our Co-IP experiments were limited by the availability of a reliable antibody specifically against MSH3 protein. In this case, the commercial antibodies worked well for immunoblotting analysis, but none were successful for the co-IP experiments. To address the reviewer's point, however, we have included a new supplementary figure, Supplementary Fig. 8, which confirms that MSH3 was present in the anti-pol β and anti-MSH2 immunoprecipitates from cells by immunoblotting with an anti-MSH3 antibody (Supplementary Fig. 8). We discussed this on Page 9, line 2, lines 5-8, lines 9-10, and lines 11-12 (red) of the revised manuscript.

c) *“The bandshift exps (Fig. 1) have been performed with both proteins. Is the bandshift reproducible then with MSH2 alone or is MSH3 in any case required?”*

Yes, MSH3 is required. MSH2-MSH3 binds stably to DNA only as a heterodimer. Indeed, MSH2 and MSH3 form only heterodimers if present together in solution. MSH3 is unstable in the absence of MSH2. In the absence of MSH3, MSH2 can form a homodimer, but it has very weak affinity for DNA.

d) "MSH2/MSH6 complex may also act on this substrate and even microsatellite instability has been ascribed to MSH2/MSH3 complex, it remains an interesting question to be addressed whether MSH2 is a scaffold protein for Pol beta, allowing formation of alternative complexes with either MSH3 or MSH6. It would be interesting to test this, but at least the authors should discuss this possibility, also in the context of the recent paper by Nakatani et al. SciRep. 2015, 5:11020."

We agree with the reviewer that this is an interesting idea and one we would like to test in the future, but with a system specifically designed for MSH6. Here we deal only with expansion. Currently, all mouse models agree that mice lacking MSH3 attenuate their expansion, while those lacking MSH6 are able to expand their repeats. Thus, MSH3 is required for expansion, and MSH6 is not. Nakatani et al. nicely demonstrates this again with a very useful cell line. As stated by those authors "Knockdown of MSH2 and MSH3, which form the MutS β heterodimer, suppressed large repeat expansions, whereas knockdown of MSH6, which forms the MutS α heterodimer with MSH2, promoted large expansions exceeding 200 repeats by compensatory increases in MSH3 and the MutS β complex". Thus, there is general agreement now in all systems that any residual MSH2-MSH6 that forms in the absence of MSH3 (siRNA in the cell experiment) is unable to support expansion. Nakatani et al. finds that knockdown of MSH6 (siRNA), however, increases the level of MSH3, which is capable of causing expansion in human cells. It seems that the cell line and mouse data agree that MSH3 is the causative agent, and MSH6 is less important in this process. We have discussed this point on Page 3, lines 18-19 (red) and have added this reference to the citation list.

3) Page 10, line 10. Along the line of the previous point, while the experiments in LoVo cells are pretty convincing, it should be better to have also a gain-of-function experiment, where LoVo are complemented with exogenous MSH2 to see whether colocalization is restored.

We appreciate the reviewer's comments. Indeed, we considered this experiment, but we were concerned that overexpression of MSH2 might lead to unforeseen artifacts. Thus, we chose to rely on LoVo cells (MSH2-deficient) as a negative control, and used two complementary approaches (by co-localization and co-immunoprecipitation) to establish that MSH2 formed a complex with pol β .

4) Page 10, line 14, Supp. Fig. 5. Actually treatment with K_2CrO_4 apparently caused a significant increase, higher than with $KBrO_3$ (longer tail in the pixel analysis panel), which was not apparent in fig. 3f. Is there a difference in the amount/kind of damages caused by these two agents?

Yes, there is a difference in the kind of damages caused by these two agents. Chromate (K_2CrO_4) predominantly induces single-stranded DNA (ssDNA) break intermediates (Messer J, Reynolds M, Stoddard L, Zhitkovich A (2006) Causes of DNA single-strand breaks during reduction of chromate by glutathione in vitro and in cells. Free Radic Biol Med 40: 1981–1992) that can directly capture pol β . On the other hand, bromate ($KBrO_3$) induces 8-oxoG (Kawanishi S, Murata M (2006) Mechanism of DNA damage induced by bromate differs from general types of oxidative stress. Toxicology 221: 172–178.) that needs to be removed by DNA glycosylases to leave an abasic site for incision by APE1. APE1 incision provides a 5'-sugar phosphate that can be bound by pol β . Results in published work verify that the frataxin protein is reduced in patient cells, and subsequently decreases the activity of DNA glycosylases bearing iron-sulfur clusters (Thierbach R., Drewes G, Fusser M, Viogt A,

Kuhlow D, Blume U, Schulz TJ, Reiche C, Glatt H, Epe B, Steinberg P and Ristow M (2010) The Friedreich's ataxia protein frataxin modulates DNA base excision repair in prokaryotes and mammals. *Biochem J.* 432(1):165-72). This then leads to the accumulation of a mismatch of A:8-oxoG that may inhibit the removal of 8-oxoGs, thereby resulting in accumulation of the oxidized base lesion.

5) Page 10, line 16-17. Here the authors make a general consideration on the possible role of these proteins in normal BER, specifically arguing that lack of colocalization of MSH2 and Pol beta reflects unrepaired oxidized bases. However, this was not proven. It is also unclear why the basal level of oxidative stress should be intrinsically higher in FRDA cells. Moreover, if the hypothesis of saturation of repair systems is true, an higher increase in 8oxoG accumulation should be observed in treated FRDA cells with respect to normal cells. This can be directly assessed by IF or 8oxoG quantification by ELISA. Lastly, it may be possible to speculate that saturation of repair systems may lead to accumulation of DNA damage, eventually converted into ssDNA breaks by incomplete repair. A neutral comet assay may also show whether there is an higher accumulation of SSBs in FRDA cells vs wild type, either untreated or after treatment.

Good point. We appreciate the opportunity to clarify. Previously published works have already established that oxidative damage is elevated in FRDA lymphoblasts that we used here (Thierbach R., Drewes G, Fusser M, Viogt A, Kuhlow D, Blume U, Schulz TJ, Reiche C, Glatt H, Epe B, Steinberg P and Ristow M (2010) The Friedreich's ataxia protein frataxin modulates DNA base excision repair in prokaryotes and mammals. *Biochem J.* 432(1):165-72). The higher level of damage appears to arise from decreased expression of the frataxin protein. The role of frataxin is to assemble iron sulfur clusters that are essential for efficient electron transfer during energy metabolism (Gomes, C.M. and Santos, R. (2013) Neurodegeneration in Friedreich's ataxia: from defective frataxin to oxidative stress. *Oxid Med Cell Longev*, 2013, 487534.). When frataxin is lost or low (as in FRDA), there is deposition of iron, which carries out Fenton chemistry and generates ROS. Oxidative damage to mitochondria is the major cause of death for human FRDA patients, and in severe cases, these iron deposits are prominent in the heart and brain of affected children. In the revised manuscript, we have added this citation. It now reads "These observations were consistent with the fact that oxidative damage is elevated in FRDA cells (Page 10, line 19)"

6) Page 11, line 5. In Supp. fig. 6a, lane 4 pol beta seems associated to HTT CAG even in untreated conditions (if I am not mistaken of course). This is consistent with previous studies (e.g. *PLoS Genet.* 2009 Dec; 5(12): e1000749), but it is not the same profile as shown in fig. 4a, lane 4. Maybe the authors could comment.

Since endogenous DNA damage occurs throughout life in every cell, some pol β was recruited to the CAG repeats in the *HTT* gene in HD patient lymphoblasts independently of exogenous DNA damage (**Supplementary Fig. 6**), as also shown previously in the striatum of HD mice (Goula, A.V. et al. (2009) Stoichiometry of base excision repair proteins correlates with increased somatic CAG instability in striatum over cerebellum in Huntington's disease transgenic mice. *PLoS Genet* 5, e1000749). We discussed this on Page 11, lines 6-9 (red) of the revised manuscript.

7) Page 13, line 2 - 4, figure 6. It appears that the loop is stabilized and the adjacent sequences protected by S1 digestion (compare 6b lanes 4,5 to 6c lanes 10, 11. This is relevant and should be discussed. Moreover, it is difficult to judge the effects of Pol beta alone, since very little digestion is seen in panel 6d. Comparing panels c and d clearly show the reduction of the prominent cleaved products at 53-55 (consistent with the lack of the short loop), but it is not clear why the authors postulate the appearance of a much larger

loop, since only a slight increase in shorter fragments (23-2) is visible, but they were present also in panel b. This should be more clearly proven, maybe by repeating the S1 experiments and titrating different amounts of MSH2/MSH3 in order to see a more clear accumulation of the short products.

We thank the reviewer for close attention to the results and insightful comments. Indeed, we agree that we did not state the results clearly enough. The entire section relevant to Fig. 6 has been re-written. In agreement with the reviewer, we do find that MSH2-MSH3 stabilizes the substrate, which was not emphasized on the first submission. In the revised manuscript, we provide a much clearer description of the results, as suggested by the reviewer. The manuscript is much improved by this change (Page 13, lines 1-25, Page 14, lines 1-6).

8) Page 13, line 13. The template strand downstream the C at position +1 is composed of a tract containing 10 CTT repeats. The products resolved on the gel show that Pol beta was able to perform strand displacement (or traverse the ss loop formed according to the hypothesis in fig. 6) copying either 1-2 or 6-7 of these repeats. There is no evidence that these will be added as extra repeats. So the word "insertion" or "added" as in the panels should be replaced by "copied". The authors may want to refer already here to the results shown in Fig. 7, with the Fen1 digestion, in support to their conclusions.

As suggested by the reviewer, the word "added" in Fig. 7b and 7c has been replaced by "copied" in the revised Fig. 7 of the revised manuscript.

9) Page 13, lines 7 - 9. Here it is written that Pol beta was able to fill the gap (lines 7-8) in the absence or presence of Pol beta (line 9)? Supplementary Figure 7 actually shows a S1 digestion experiment where no major loop formation can be seen. However it is not immediately clear how gap-filling activity can be judged from this experiment. The authors may want to perform the same experiment as shown in Fig. 7 of the paper (DNA synthesis) also on this random substrate to show precise gap filling by Pol beta.

We have corrected the wording error. Supplementary Fig. 7 showed S1 cleavage on the 1 nt-gapped random sequence substrate, which does not have any secondary structure (Supplementary Fig. 7). The nuclease cleavage resulted in only a single product with 53 nt in the absence of pol β or MSH2-MSH3, in the presence of pol β or MSH2-MSH3, and in the presence of both proteins (Supplementary Fig. 7). Thus, the production of the single 53 nt product from S1 cleavage indicated the absence or elimination of secondary structures (Fig. 6e,k). We have corrected the wording error and rewrote the text on Page 13, lines 22-24 of the revised manuscript in red.

10) Page 14, line 10. "no flaps were observed". It is not clear what the authors meant. Flaps are detected by Fen1 cleavage. In the absence of Fen1 no cleavage was observed, which does not mean there were no flaps. Maybe I am missing something here, and if so I apologize, but if the authors suppose that Fen1 itself promotes Flap formation, they should prove in some independent way that no flaps were present when Pol beta was included. Also Page 14, line 16 "MSH2/MSH3 alone was ineffective at generating a flap". This is contradictory to the previous results of the S1 digestion. MSH2-3 likely generate a long flap, but cannot cleave it in the absence of Fen1. Maybe this can be rectified by writing "...alone cannot generate cleavage products..."

We thank the reviewer for careful readings. We agree and we have changed the wording. As the reviewer suggested, we changed this to "MSH2-MSH3 alone cannot cut the DNA" (Fig. 7h, lane 7)". We modified the sentences on Page 15, line 5 and lines 11-12 of the revised manuscript.

Minor points

1) Page 5, line 16. "...for the first time...". It is always better to avoid too firm statements. Adding "to the best of our knowledge" is suggested.

We appreciate the reviewer's helpful suggestion. We modified this sentence to read "Here, to the best of our knowledge, we provide the first direct evidence for a molecular crosstalk mechanism, in which MSH2-MSH3 is used as a component of the BER machinery to cause expansion" on Page 5, lines 18-20 of the revised manuscript.

2) In the Introduction or Discussion, the authors may want to cite also Nakatani et al. *SciRep.* 2015, 5:11020 where *msh2/msh3* knock down suppresses repeat expansion in cells (see also major point 2 for a possible specular role of MSH6).

According to the reviewer's suggestion, the reference has been cited as "reference 26" in the revised manuscript (Page 31, lines 10-12, red).

3) In many places, the authors state that they pre-treated the substrate with APE1 to make the incision. However, such incision should be formally proven by running the substrate on a denaturing gel in order to see the accumulation of the 5'end labelled strand cleaved product. This is shown only in figure 7 b. At least the authors should specifically refer to such figure as a demonstration that APE1 can indeed cleave the synthetic abasic site.

We agree. In the revision, we have referred to Fig. 7b. The text on Page 14 now reads: APE1 cleavage produced the expected incision product (see Fig. 7b, lane 2) (Page 14, lines 11-12 in red).

4) Page 12, line 25. panels 6d and 6f represent schematic diagrams of the substrates. Maybe the authors meant to refer to the fragment analysis shown in figure 5? Then the authors should say so.

Panel 6f is the schematic diagram that is below each gel result. Fig. 6d is a result. We have clarified the designations in the revised text. The text on Page 12 now reads "The schematic diagram of the results of each substrate product is below each panel in (Fig. 6f,I)." (Page 12, lines 23-24)

5) 80 references are probably too many for a research paper. I would suggest to reduce them to less than 70.

We appreciate the reviewer's helpful suggestion. The number of references has been cut down to 70.

Reviewer #2 (Remarks to the Author):

This manuscript describes an interesting and mostly well-executed set of experiments that demonstrate that MutS β , a component of the mismatch repair and Pol β , a component the base excision repair pathways co-localize in vivo in response to oxidative damage and interact in vitro in such a way as to generate products that could result in expansions of GAA and CAG repeats in vivo. Their interaction in vitro increases the processivity of the normally not very processive Pol β enzyme. The authors' data suggest that in vivo this could reduce the generation of products that result in deletions and increase the generation of products that could in principle lead to expansions. This data provides a plausible explanation of the key steps of the mutational mechanism that gives rise to the Repeat Expansion Diseases.

1. Most of my concerns are relatively minor involving either confusing/incomplete descriptions or the use of overly broad statements that are not fully supported by the data. However, some data is unconvincing or open to other interpretation. For example, MSH2 and Pol β are said to co-localize in data shown Fig. 3f. However, the data provided show very little clear evidence of that, with both proteins showing a very wide distribution.

We understand the reviewer's concern. As expected, there is wide-spread staining of both proteins, which is unavoidable for two constitutive proteins that reside throughout the cells. However, that is why we use pixel analysis, where no interpretation is required on our part. Pixel analysis is a quantitative measure that is automatically collected by the microscope--every pixel in the image is recorded in every color channel, i.e., the same pixel is counted in the green channel as green, in the red channel as red, and the overlap in identical images tell us if they have both red and green. There is no guesswork. We were simply reporting the number of pixels that have both red and green as recorded by the microscope, even if foci were not easily distinguished. This is a standard analysis by Zeiss software as well as other software packages, and we have begun (as have others) to use pixel analysis for every co-localization experiment to take the guesswork out of the results. Thus, we stand by these measurements. These data can be downloaded for the reviewer. Supplementary Figure S5 is provided for the reader who is not familiar with the analysis.

2. In addition, on Pg. 13 lines 4-12, it is claimed that the behavior of Pol β on a 1nt-gapped substrate containing a repeat is "remarkable" in that "Pol β was unable to traverse the repeat loop on the template" generating a large single stranded region that was not seen on a template that did not contain the repeat (Fig. 6 and Supplementary Fig. 7). This result is very difficult to understand in terms of the authors' explanation since it is unclear how the long regions of the repeat would become unpaired during the reaction given that Pol β is not very processive and the reaction contains no accessory proteins that may facilitate this unpairing. An alternative explanation is that with a poorly processive polymerase like Pol β , strand dissociation there is a lot of strand dissociation. However, only on a repeat-containing template is there efficient strand-slippage and repriming from the slipped position. Could it be that the large region of S1 sensitivity seen on the repeat template does not reflect a single large loop out as the authors suggest, but a series of small loop outs resulting from strand slippage and repriming at multiple places in the repeat tract. When the processivity of Pol β is increased in the presence of MutS β , then strand slippage is less likely to occur (while the likelihood of strand-displacement increases). Thus, it may be that the data in Fig. 6 reflects the possibility of strand slippage on a template containing repeats rather than Pol β being unable to traverse a loop generated on the template, a perhaps subtle but important difference that would not be so much remarkable, but consistent with expectations.

We agree that MSH2-MSH3 increases the processivity of pol β . We have re-written the section on Page 13 and Page 14 (Page 13, lines 12-25, Page 14, line 1 (red)) as follows:

"To test whether the single-stranded region was a gap or a loop, we further determined the S1 cleavage pattern with or without pol β synthesis (**Fig. 6d and Supplementary Fig. 9b**). We found that pol β protein alone without nucleotides mainly led to the same S1 cleavage bands between +53 and +48 as those in the absence of repair proteins (compare **Supplementary Fig. 9b,c** with **Fig. 6b,f**). Thus, the ladder of bands between +53 and +23 required pol β DNA synthesis (**Fig. 6d,f**). The results indicated that pol β synthesis facilitated the formation of a large CTT loop rather than a gap on the template strand. This further suggests that pol β synthesis displaced the downstream strand, thereby promoting repeat slippage and the formation of a large (CTT)₁₂ loop on the template strand (**Fig. 6d,f**). However, when MSH2-MSH3 and pol β were added together, S1 cleavage only resulted in a single 53 nt product, indicating that loop on the template strand was eliminated (**Fig. 6e,f**). Consistent with this, S1 cleavage on the 1 nt-gapped random sequence substrate, which

does not have any secondary structure, resulted in only a single product with 53 nt (**Supplementary Fig. 7**). Thus, the results indicated that the presence of both pol β synthesis and MSH2-MSH3 eliminated the loop on the template strand.”

3. *The remaining points are addressed more or less in the order in which they appear in the manuscript.*

1) *The title of the manuscript should be amended to include the words "in vitro" since no data are presented proving that the mechanism delineated in vitro is in fact what is responsible for expansion in vivo.*

We prefer to leave it as it is because these *in vitro* experiments follow our previous *in vivo* experiments showing that MSH2 and OGG1 act in the same pathway to cause expansion *in vivo*. The results here provide a deeper understanding of the mechanism by which they interact.

2) *Pg. 6 (and the abstract). The phrase beginning with "Since genetic interactions of MSH2-MSH3, OGG1, NEIL1 and Pol β cause expansions in mice" should be rephrased. Each of those proteins have been shown to promote expansions however, strictly speaking no interactions genetic or otherwise have until now been demonstrated.*

According to the reviewer’s suggestion, we rephrased the sentence as “Since MSH2-MSH3, OGG1, NEIL1, and pol β have been implicated in causing expansions in mice, we asked whether there are interactions between the MMR and BER machinery.” We have modified the sentence on Page 6, lines 7-9 in red.

3) *Pg. 6. It is stated that there was little evidence for a direct interaction of MSH2-MSH3 with OGG1 on templates with an oxidized base. However, the data for this (Supplementary Fig. 2) does seem to show some evidence of interaction. Unfortunately since the experiment used a different methodology from that used to study the interaction MSH2-MSH3 with Pol β , a direct comparison is difficult.*

We appreciate the reviewer’s careful reading and constructive comments. In the experiment, the interaction between MSH2-MSH3 and OGG1 on templates with an oxidized lesion was probed using gel filtration chromatography. OGG1 has a molecular weight of around 40kD while MSH2-MSH3 is around 200kD. They would co-migrate only if they formed a physical complex. Our results clearly demonstrated that the OGG1-DNA peak is resolved from the MSH2-MSH3-DNA peak. In addition, in the fractions containing unbound OGG1, MSH3 was not detected indicating that the two proteins failed to interact with each other. The leading edge of the OGG1-DNA peak does elute with the trailing edge of the MSH2-MSH3-DNA peak in a few fractions.

4) *Why does the CAG substrate binds less well to MutS β than the GAA template in Fig. 1b but not Fig. 1c? Also, it is not mentioned in the text but it is apparent from Fig. 1b, that MutS β also improves APE1 binding (again more to the GAA template than the CAG template). The nature of this binding should be addressed since it does not seem to involve an interaction of MutS β with APE1 that is stable enough to survive electrophoresis.*

There is no difference; it reflected only a different batch of radiolabel. We have replaced Fig. 1b with a new one that showed no difference in MutS β binding to CAG and GAA repeat substrates in the revised manuscript.

Regarding APE1 substrate binding, it is unlikely the lower band was APE1•MSH2-MSH3•DNA complex. MSH2-MSH3 is a large protein complex (230kD) and the band was also detected in the binding reactions containing MSH2-MSH3 and the DNA substrate without the presence of APE1 (Fig. 1c, lanes 3 and 8). In Fig. 1c, the same band is observed

when only MSH2-MSH3 is present, suggesting that there may have been a small amount of truncated oligonucleotide whose end will bind to MSH2-MSH3. We have added this point to the Figure legend of Fig. 1 of the revised manuscript (Page 34, lines 20-22).

5) Could the authors comment on the fact that while it is true that the protection of the substrate by Pol β and MutS β is more dramatic on one side of the nick, there is a subtle but distinct change in the DNase I footprint seen downstream of the repeat, at least in the case of the GAA repeat (Supplementary Fig. 3)?

Yes, our results are consistent on this point. Some wiggling of the proteins is likely to create shadows on the footprint of the other strand in practice, but the only real protection is obvious in the gels. According to the reviewer's comment, however, the binding sites of pol β and MSH2-MSH3 were indicated in Supplementary Fig. 3 and better described on Page 8, lines 9-12 of the revised manuscript.

6) The authors should cite evidence for the statement that oxidative damage is elevated in FRDA patient cells to support their contention that the absence of increased colocalization of Pol β and MutS β in response to oxidative damage in patient cells reflects the elevated background levels of oxidative damage in these cells. They should also provide evidence that this same effect is seen in lines established from other FRDA patients.

Good point. Previously published works have already established that oxidative damage is elevated in FRDA lymphoblasts that we use here (Thierbach R., Drewes G, Fusser M, Viogt A, Kuhlow D, Blume U, Schulz TJ, Reiche C, Glatt H, Epe B, Steinberg P and Ristow M (2010) The Friedreich's ataxia protein frataxin modulates DNA base excision repair in prokaryotes and mammals. *Biochem J.* 432(1):165-72). The higher level of damage appears to arise from decreased expression of the frataxin protein. The role of frataxin is to assemble iron sulfur clusters that are essential for efficient electron transfer during energy metabolism (Gomes, C.M. and Santos, R. (2013) Neurodegeneration in Friedreich's ataxia: from defective frataxin to oxidative stress. *Oxid Med Cell Longev*, 2013, 487534). When frataxin is lost or low (as in FRDA), there is deposition of iron, which carries out Fenton chemistry and generates ROS. Oxidative damage to mitochondria is the major cause of death for human FRDA patients, and in severe cases, these iron deposits are prominent in the heart and brain of affected children. In the revised manuscript, we have added this citation. It now reads "These observations were consistent with the fact that oxidative damage is elevated in FRDA cells " (Page 10, line 19).

7) Pg 11, line 16. A little more detail at this point as to how the experiment was done would make this paragraph easier to follow.

We would be happy to provide more explanation. The repaired products were pulled down with avidin beads after the termination of the BER reaction. The repaired strands were subsequently separated from their biotinylated template strands by incubating with NaOH. The separated repaired strands were subsequently amplified by using 5'-FAM labeled primers and resolved by high-resolution capillary electrophoresis to define their length. As the reviewer suggested, we have included a more detailed description of the method in the legend of Fig. 5 of the revised manuscript in red (Page 37, lines 10-14).

8) Pg. 12, lines 24-26. This section is difficult to follow and may contain errors.

We thank the reviewer for pointing out the error. The text of the section has been re-written according to the reviewer's suggestion. (Page 13, lines 8-25, Page 14, lines 1-6 in red)

9) Pg. 13, line 1-2. *It is stated that addition of MutS β has little effect on the S1 digestion of the template in the absence of Pol β , however it seems clear that MutS β is protecting the region from further S1 digestion.*

We also thank the reviewer for close attention to the results and insightful comments. Indeed, we agree that we did not state the results clearly enough. The entire section relevant to Fig. 6 has been re-written. In agreement with the reviewer, we do find that MSH2-MSH3 stabilizes the substrate, which was not emphasized on the first submission. In the revised manuscript, we provide a much clearer description of the results, as suggested by the reviewer. The manuscript is much improved by this change (see Page 13, lines 3-6 in red)

10) Pg. 13, line 6 (as well as the abstract (Pg. 2, line 6)). *The phrase "consistent with a deletion" is too imprecise. It is more accurate to say that this situation could ultimately result in the production of a deletion.*

This paragraph has been re-written according to the reviewer's suggestions (Page 13, lines 1-25, Page 14, lines 1-6).

11) Pg. 13, line 15. *Strictly speaking the experiments in this section do not confirm the results of the previous experiments. Rather they are an evaluation of what is happening on the other strand.*

The wording has been changed to "We then tested whether MSH2-MSH3 could promote expansion by stimulating pol β DNA synthesis (**Fig. 7**)". (Page 14, lines 8-9 in red)

12) Pg. 13, line 18 (as well as the abstract (Pg. 2, line 6) and Pg 15, line 9. *It is not correct to say that Pol β failed to complete gap-filling synthesis in the absence of MutS β , but rather that synthesis was less processive.*

We have changed the wording to state that pol β was less processive in the absence of MSH2-MSH3 on Page 14, line 12-13 of the revised manuscript in red.

13) Pg. 15, line 18. *The details of the "toxic oxidation cycle" model are not well described. Additionally, since the term "toxic oxidation cycle" is sometimes used to describe a somewhat different model for the role of oxidative damage in repeat expansion, it might be better to come up with a different name for the authors' model.*

We first proposed the "toxic oxidation cycle" for expansion, which was reported in Nature (Kovtun, I.V. et al (2007) OGG1 initiates age-dependent CAG trinucleotide expansion in somatic cells. *Nature* **447**, 447-52). The steps were well delineated and defined, and we must refer to our original model, as we are testing the steps we outlined. We have cited it to avoid any confusion.

14) Figure 8. *The arrow indicating the site of the "trans" acting endonuclease in the model should be placed further 5' on the bottom strand shown in the right hand panel so that repair of the nick would entail replication of the loop introduced on the top strand during flap displacement synthesis.*

We appreciate the reviewer's careful reading. The arrow indicating "Endonuclease in trans" has been moved in Fig. 8 of the revised manuscript as suggested by the reviewer.

Reviewers' comments:

Reviewer #1 (Remarks to the Author):

In this revised manuscript, the authors better corroborated their finding that a molecular cross-talk exists between BER and MMR, through the action of MSH2/MSH3, during TNR expansion. The data presented open novel perspectives and provide interesting new testable hypotheses. Their results have been obtained with an appropriate technology. Experiments are well performed and controlled and data are appropriately shown and discussed.

The authors have satisfactorily addressed my previous concerns. The text is much more readable and it is also clearer for the non specialist. Issues about experimental data and/or results interpretations have been clarified. References have been adjusted. I have no further remarks.

Reviewer #2 (Remarks to the Author):

Many issues from the first review were not properly addressed. In addition, some of the revisions have raised new issues.

About the points made in my original review:

1) While the authors are technically correct that Pixel Analysis shows that MSH2 is frequently found in the same part of the nucleoplasm as Pol β the authors overstate the meaning of this finding and the data is semi-quantitative at best, not quantitative as stated in line 210. The ZEN software does provides the means of making quantitative measurements with associated statistics, however the authors do not make use of this feature.

In addition, the resolution of Fig. 3 is inadequate and many important experimental details are either missing or incorrect.

- Line 217. Was the optical section really 0.1 μm and not perhaps 1 μm ?
- Lines 626-630. The antibodies used are incompletely described. They were Alexa antibodies presumably? Proper catalog names should be provided along with appropriate catalog numbers.
- Line 632: The optical sectioning cannot be 90 μm . Perhaps the authors are referring to the pinhole size? Instead of this value the authors should use the use the theoretical "optical section" or "slice thickness" reported by ZEN, with the appropriate hardware configuration loaded while using the same microscope, applying the "ReUse" function on the previously acquired images.
- It is also customary to report the number of AU (Airy Units), which is the normalized pinhole size that produced such an optical section.
- Other imaging conditions, some reported by Zen software, should be included in the paper:
 - o Lasers and filters used for each color. Someone trying to interpret data herein or reproduce results could assume these settings, but better to avoid ambiguity.
 - o Pixel size (on the sample). This could be important for interpretation of any kind of quantitative pixel analysis.
- Line 634: details about how the pixel analysis was done would be more appropriate here than in the legend for Fig. S4. This should include a description of what criteria were used to determine the

thresholds for the 2 channels. Critical examination of the data in Fig. 3f leads me to think that the level of "co-localization" may not be very different in treated and untreated cells, it may simply be that oxidative damage increases the levels of the proteins above the threshold chosen.

Fig S5 is unnecessary. The figure is a very slightly modified version of a figure in "Colocalization Analysis in AIM and ZEN: A Carl Zeiss How-To Guide", a document prepared by Carl Zeiss Microscopy, while the legend is taken from that figure practically verbatim. The authors could simply the appropriate references.

2) In the original manuscript it was stated that "Pol β was unable to traverse the repeat loop on the template generating a large single stranded region that was not seen on a template that did not contain the repeat (Fig. 6 and Supplementary Fig. 7)." I pointed out in my review that from their data it is impossible to distinguish such a loop from the perhaps more likely possibility that this reflects a series of small loops generated by repeated strand-slippage. The authors responded as follows: "To test whether the single-stranded region was a gap or a loop, we further determined the S1 cleavage pattern with or without pol β synthesis (Fig. 6d and Supplementary Fig. 9b). We found that pol β protein alone without nucleotides mainly led to the same S1 cleavage bands between +53 and +48 as those in the absence of repair proteins (compare Supplementary Fig. 9b,c with Fig. 6b,f). Thus, the ladder of bands between +53 and +23 required pol β DNA synthesis (Fig. 6d,f). The results indicated that pol β synthesis facilitated the formation of a large CTT loop rather than a gap on the template strand. This further suggests that pol β synthesis displaced the downstream strand, thereby promoting repeat slippage and the formation of a large (CTT)₁₂ loop on the template strand (Fig. 6d,f). However, when MSH2-MSH3 and pol β were added together, S1 cleavage only resulted in a single 53 nt product, indicating that loop on the template strand was eliminated (Fig. 6e,f). Consistent with this, S1 cleavage on the 1 nt-gapped random sequence substrate, which does not have any secondary structure, resulted in only a single product with 53 nt (Supplementary Fig. 7). Thus, the results indicated that the presence of both pol β synthesis and MSH2-MSH3 eliminated the loop on the template strand."

It is not unclear to me how the experiments without dNTPs address the question of repeated strand-slippage vs the generation of a large loop. My original point remains unanswered.

3) In my original review I pointed out that there does seem to be some interaction between OGG1 and MSH2-MSH3 as evidenced by their presence in the same eluted fraction. The authors' responded by saying:

"Our results clearly demonstrated that the OGG1-DNA peak is resolved from the MSH2-MSH3-DNA peak. In addition, in the fractions containing unbound OGG1, MSH3 was not detected indicating that the two proteins failed to interact with each other. The leading edge of the OGG1-DNA peak does elute with the trailing edge of the MSH2-MSH3-DNA peak in a few fractions."

Since some eluted fractions clearly contain both sets of proteins, I don't think that the authors explain clearly why they conclude that OGG1 and MSH2-MSH3 don't interact. The figure legend (Supplementary Figure 2) simply states that the DNA protein complexes were resolved by column chromatography (no details for this chromatography are provided) and that eluted "Fractions were

resolved using band shift analysis". This needs better explanation. Do the authors mean that a product containing both OGG1 and MSH2-MSH3 bound to DNA would elute at a very different position on the column? That there were no products corresponding to both OGG1 and MSH2-MSH3 bound to the same substrate in an EMSA? The absence of MSH2-MSH3 in the peak that contains unbound OGG1 may reflect the fact that all of the MSH2-MSH3 is bound to the hairpin, since no free MSH2-MSH3 is seen.

4) Both reviewer #1 and myself commented on the fact that MutS β also seems to improve APE1 binding (Fig. 1b).

The authors claim that the band that with the same mobility as APE1 was in fact the result of binding of MSH2-MSH3 to the ends of a truncated oligonucleotide as evidenced by the presence of a similar sized band in the reactions with MSH2-MSH3 alone. This explanation seems unlikely to me for 3 reasons

- 1) the band seen with MSH2-MSH3 alone is much less abundant than the band seen in presence of APE1
- 2) it is also much broader
- 3) it is the protein not the DNA that is the major determinant of the mobility shifts in this sort of assay. Thus binding to truncated oligonucleotide seems an unlikely explanation for this band. The authors could easily resolve the question by using full length and truncated probes.

5) The phrase "Genetic interactions" has been rephrased in the body of the manuscript but not the first sentence of the abstract.

6) Reviewer #1 also said: "MSH2/MSH6 complex may also act on this substrate and even microsatellite instability has been ascribed to MSH2/MSH3 complex, it remains an interesting question to be addressed whether MSH2 is a scaffold protein for Pol beta, allowing formation of alternative complexes with either MSH3 or MSH6. It would be interesting to test this, but at least the authors should discuss this possibility, also in the context of the recent paper by Nakatani et al. SciRep. 2015, 5:11020."

The authors' responded by stating that MSH6 is less important than MSH3 for expansion, the implication being that there is no need to test whether MSH2-MSH6 acts in a similar way to MSH2-MSH3 with respect to Pol β in the in vitro assay used. However, since MSH6 and MSH3 share many common properties, it may well do so. If it did not, it would strengthen the argument that MSH2-MSH3 acts to promote repeat expansion in the manner they suggest. This should at least be discussed.

Additional comments (my apologies if some of these comments were not provided during the first review).

1) The phrase: Indeed, at TNRs, transgenic mice lacking 8-oxo-guanine glycosylase (OGG1), NEIL1, pol β with normal activity, and XPA inhibit expansion" should be rephrased. The mice are not inhibiting expansion. The lack of those proteins is preventing expansions from happening.

2) With respect to Fig. 1, it is stated that MSH2-MSH3 significantly stimulated Pol β loading. It is unclear to me that the authors have evidence that Pol β loading is improved but rather in the presence of both sets of proteins there is more substrate binding. Without specifically examining the levels of free and bound Pol β using something like antibody supershifts or IP, the authors cannot speak to whether Pol β loading is increased or not.

3) In the text added to Pg 8 it is stated that there "There was little protection from either protein on the downstream strand of the (GAA)₂₀ and (CAG)₂₀ substrates (Fig. 2b,e). However, some protection from both proteins was detected on the upstream strand of the substrates (Supplementary Fig. 3)." I presume that the authors mean that there is protection of the bottom strand but not of the top strand.

4) On Pg 12 it is stated that MSH2-MSH3 suppressed deletion and promoted expansion. How do the authors exclude the possibility that when Pol β synthesis does not result in deletions there is more opportunity for strand-displacement that results in expansions, i.e., that increased expansions are an indirect effect of reduced deletions? Also, the major products of GAA repair are expansions even in the absence of MSH2-MSH3. The authors should discuss why MSH2-MSH3 is essential for expansion in most mouse models of the repeat expansion disorders, yet is not obligatory in vitro at least for GAA. Does this relate to the rather odd data reported for the effect of MMR mutations in the FRDA (GAA-repeat) mouse?

5) (Minor point) Can the authors really be sure that the peak labeled CAG₁₈ is really 18 and not 17? The spacing between the peak labeled CAG₁₉ and CAG₁₈ seems a little large.

6) Pg. 15, text has been added to the effect that no flap cleavage was seen in the absence of FEN1 (line 347; Fig. 7h, lane 1-3) and later that MSH2-MSH3 alone cannot cut the DNA (line 354; Fig. 7h, lane 7). However, none of those lanes contain both MSH2-MSH3 and Pol β but not FEN1. Without Pol β , there would be no flaps to cut.

Author response to reviewer comments

Reviewer #1 (Remarks to the Author):

In this revised manuscript, the authors better corroborated their finding that a molecular cross-talk exists between BER and MMR, through the action of MSH2/MSH3, during TNR expansion. The data presented open novel perspectives and provide interesting new testable hypotheses. Their results have been obtained with an appropriate technology. Experiments are well performed and controlled and data are appropriately shown and discussed.

The authors have satisfactorily addressed my previous concerns. The text is much more readable and it is also clearer for the non specialist. Issues about experimental data and/or results interpretations have been clarified. References have been adjusted. I have no further remarks.

Reviewer #2 (Remarks to the Author):

(1) While the authors are technically correct that Pixel Analysis shows that MSH2 is frequently found in the same part of the nucleoplasm as Pol β the authors overstate the meaning of this finding and the data is semi-quantitative at best, not quantitative as stated in line 210. The ZEN software does provide the means of making quantitative measurements with associated statistics, however the authors do not make use of this feature.

The reviewer comments that the Pixel Analysis correctly shows that MSH2 is frequently found in the same part of the nucleoplasm as Pol β , and complemented the immunoprecipitation (IP) experiments. However, the reviewer cautions that we overstated the quantitative nature of these data and has requested inclusion of more details of the pixel analysis. We agree, and we have modified the text accordingly.

In the revised manuscript, we have re-written the section to remove overstatements of the results. (1) We explicitly limit the scope of the discussion to be supportive of the IP experiments (Page 10, lines 212). (2) We have revised the text to include experimental details asked for by the reviewer, as outlined below.

(a) Line 217. Optical sections.

We thank the reviewer for attention to detail, and indeed, the optical sections were taken every 1 μm (Page 10, line 219).

(b) Lines 626-630. The antibodies used are incompletely described.

Due to the strict word limit, we confined a detailed description of the antibodies to the methods section. To avoid confusion, we have included additional text to direct the reader to the methods. Primary antibodies were: rabbit polyclonal anti-MSH2 antibody (1:200, (N20) sc494, Santa Cruz Biotechnology, Dallas, TX), mouse monoclonal anti-DNA polymerase β antibody (1:200, #ab1831, Abcam, Cambridge, MA). Secondary antibodies were: donkey anti-rabbit 568 (donkey anti-rabbit IgG (H+L) secondary antibody, Alexa Fluor 568 conjugate 1:250, #A10042, Life Technologies, Carlsbad, CA) and goat anti-mouse 488 (goat anti-mouse IgG (H+L) secondary antibody, Alexa Fluor 488 conjugate (1:250, #A11001, Life Technologies, Carlsbad, CA) (Page 25, 595-611).

(c) Line 632: The optical sections were 1 μm , which has been corrected in the text (Page 10, line 219).

(d) It is also customary to report the number of AU (Airy Units), which is the normalized pinhole size that produced such an optical section.

All of the confocal imaging utilized a pinhole set to provide 1AU (0.7 μm sections using Plan apochromat 100x/1.4 oil DIC objective). This information has been included in the Methods (Page 25, lines 608-609).

(e) Other imaging conditions, some reported by Zen software, should be included in the paper: Lasers and filters used for each color. Someone trying to interpret data herein or reproduce results could assume these settings, but better to avoid ambiguity.

The following has been included in the Methods (Page 25, lines 609-611).

- Lasers: Argon (458,488,514nm), DPSS 5610-10 (561nm), HeNe (633nm), Diode 405-1 (405nm)
- Filters: MBS 458/514 (visible light), MBS 405 (invisible wavelengths)
- Pixel size (on the sample): 514x514 pixels per frame

(f) Line 634: details about how the pixel analysis was done would be more appropriate here than in the legend for Fig. S4.

(g) Fig. S5 is unnecessary.

(I assume that the reviewer is referring to Fig. S4).

We agree that Fig. S4 is unnecessary. However, we have exceeded the word limit and have been asked by the editors to reduce the main text. Thus, we have left in Fig. S4, and refer the reader to the legend (Supplementary information, Page 4) and to the methods (Page 25, lines 607-611). However, if the reviewer feels strongly that we should remove it, we will do so. Please let us know.

(2) "In the original manuscript it was stated that "Pol β was unable to traverse the repeat loop on the template generating a large single stranded region that was not seen on a template that did not contain the repeat (Fig. 6 and Supplementary Fig. 7)." The reviewer comments that the data presented did not distinguish a large loop from a series of small loops repeated during strand-slippage. The reviewer did not feel that we answered the questions thoroughly enough, and it was unclear how the as added experiments without dNTPs addressed the question.

We agree that the data show only that a large single stranded region was formed. Thus, we re-wrote the text to include both possibilities: either a large loop or a series of smaller ones. We agree that the "no dNTP" control demonstrates only that the looped regions required active polymerase (Page 13, 303-312, Page 14, line 313).

(3) In my original review I pointed out that there does seem to be some interaction between OGG1 and MSH2-MSH3...as evidenced by their presence in the same eluted fraction...I don't think that the authors explain clearly why they conclude that OGG1 and MSH2-MSH3 don't interact.... no details for this chromatography are provided)...This needs better explanation.

We apologize that the section was unclearly written and details were not sufficient—we agree. we have revised the legend of Supplementary Fig. 2 to explicitly discuss the interpretation of the results (described below)(Supplementary information, Page 2). The complete antibody description is included in the methods section, which is also indicated in the revised Supplementary information (Supplementary method, Page 13).

The legend now reads:

Supplementary Figure 2 illustrates the chromatographic resolution on a Sephacryl S-100 matrix of a 1:1:1 mixture of OGG1, MSH2-MSH3 and an 80-mer oligonucleotide duplex containing one 8-oxo-G residue (in red).

CAG-duplex (80)-

5'CAAGCACGTTGACTACCGTCCAGCAGCAGCAGCAGCAGCAGCAGCAGCAGCAGCAGCAGCAG
CAG-TTTGAGGCAGAGTCCGAACAC-3'

Column fractions along the elution profile were probed using antibodies for human OGG1 and MSH2-MSH3 in each fraction, and are shown below the elution curve. The OGG1 protein is a small protein (39 kD), while MSH2-MSH3 is roughly 230 kD. They should not co-elute due to the size differences. As indicated by the elution profile and the western analysis, no unbound OGG1 overlaps with MSH2-MSH3, indicating that the proteins do not associate off DNA. A product containing both OGG1 and MSH2-MSH3 bound to DNA would be larger than the MSH2-MSH3-DNA complex, and would be the fastest migrating species in the elution profile. However, as detected in the blots, no OGG1 or MSH2-MSH3 co-elutes with the fastest migrating species in the elution fractions. Thus, the two proteins do not appear to interact, but are observed in one fraction where there is modest overlap of elution fractions. We have included these details in the Figure S2 legend (Supplementary information, Page 2).

(4) Both reviewer #1 and myself commented on the fact that MutS β also seems to improve APE1 binding (Fig. 1b). The authors claim that the band that with the same mobility as APE1 was in fact the result of binding of MSH2-MSH3 to the ends of a truncated oligonucleotide as evidenced by the presence of a similar sized band in the reactions with MSH2-MSH3 alone. This explanation seems unlikely ...

We also like the idea that MSH2-MSH3 might stimulate APE1 binding, but based on the data, we could not claim it. We based our original statement on the observations that the two bands in MSH2-MSH3-APE1 lane (Fig. 1b, lane 4) are identical to those of MSH2-MSH3 in the absence of APE1 (see Fig. c, lane 3). As suggested by the reviewer, we repeated the binding of MSH2-MSH3 with a purified DNA substrate, but the pattern in Fig. 1 was the same. Furthermore, if MSH2-MSH3 improved APE1 binding, then both proteins would need to be present in the complex, and we would expect that the upper complex would be larger than MSH2-MSH3-DNA, which is not the case (compare Fig. 1b, lane 4, to 1c, lane 3). Taken together, we favor the reviewer's idea that a small amount of truncated MSH2-MSH3 proteins bound to the DNA substrate resulting in the lower band. We discussed this point in the legend of Fig. 1 in the revised manuscript (Page 37, lines 897-899).

(5) The phrase "Genetic interactions" has been rephrased in the body of the manuscript but not the first sentence of the abstract.

We have rephrased the first sentence of the abstract according to the reviewer's comments.

(6) Reviewer #1 also said: "MSH2/MSH6 complex may also act on this substrate and even microsatellite instability has been ascribed to MSH2/MSH3 complex, it remains an interesting question to be addressed whether MSH2 is a scaffold protein for Pol beta, allowing formation of alternative complexes with either MSH3 or MSH6. It would be interesting to test this, but at least the authors should discuss this possibility, also in the context of the recent paper by Nakatani et al. SciRep. 2015, 5:11020." This should at least be discussed.

We have re-written the discussion to bring up the possibility that MSH6 might also be a scaffold for pol β . It now reads: "We have yet to test whether MSH2-MSH6 acts as a scaffold for pol β , but it is also possible, given their structural similarity, that MSH2-MSH6 participates in crosstalk by modifying pol β -MSH2-MSH3-mediated TNR instability." (Page 18, lines 416-418).

New comments

(1) The phrase: Indeed, at TNRs, transgenic mice lacking 8-oxo-guanine glycosylase (OGG1), NEIL1, pol β with normal activity, and XPA inhibit expansion" should be rephrased. The mice are not inhibiting expansion. The lack of those proteins is preventing expansions from happening.

We corrected the sentence by rephrasing: "...expansion is reduced in transgenic mice lacking 8-oxo-guanine glycosylase (OGG1), NEIL1, pol β with normal activity, and XPA, strongly supporting the notion..." (Page 5, lines 85-87).

(2) With respect to Fig. 1, it is stated that MSH2-MSH3 significantly stimulated Pol β loading. It is unclear to me that the authors have evidence that Pol β loading is improved but rather in the presence of both sets of proteins there is more substrate binding. Without specifically examining the levels of free and bound Pol β using something like antibody supershifts or IP, the authors cannot speak to whether Pol β loading is increased or not.

We agree with the reviewer about this point. We removed the sentence according to the reviewer's comment.

(3) In the text added to Pg 8 it is stated that there "There was little protection from either protein on the downstream strand of the (GAA)₂₀ and (CAG)₂₀ substrates (Fig. 2b,e). However, some protection from both proteins was detected on the upstream strand of the substrates (Supplementary Fig. 3)." I presume that the authors mean that there is protection of the bottom strand but not of the top strand.

We agree that the wording "upstream and downstream" is not clear. We have revised the text for better clarification. The text now reads, "There was little protection from either protein on the damaged strand (top strand) of the (GAA)₂₀ and (CAG)₂₀ substrates (Fig. 2b,e). However, protection by both proteins was detected on the template strand (bottom) (Supplementary Fig. 3). We modified the sentences on Page 8, lines 173-176.

(4) On Pg 12 it is stated that MSH2-MSH3 suppressed deletion and promoted expansion. How do the authors exclude the possibility that when Pol β synthesis does not result in deletions there is more opportunity for strand-displacement that results in expansions, i.e., that increased expansions are an indirect effect of reduced deletions?

We are honestly confused by the distinction. The reviewer has identified exactly the important points. Deletions occur only when MSH2-MSH3 is absent since pol β does not copy the TNR repeats. When MSH2-MSH3 is present, no deletion occurs (deletions are suppressed) and it promotes strand displacement synthesis causing the expansions. We modified the sentences on Page 12, lines 277-280.

(5) Also, the major products of GAA repair are expansions even in the absence of MSH2-MSH3. The authors should discuss why MSH2-MSH3 is essential for expansion in most mouse models of the repeat expansion disorders, yet is not obligatory in vitro at least for GAA.

Good point. Indeed, we believe our results explain, at least in part, why there are expansions upon loss of MSH2-MSH3 in the FRDA mice. The main point of the figure is that the prominent deletion bands is diminished by addition of MSH2-MSH3, and expansions are favored. The general trend for deletion suppression and expansion promotion on GAA and CAG are similar. However, since expansion is determined by the difference between the nucleotides added by pol β synthesis and those removed by FEN1, the need for MSH2-MSH3 to increase processivity of the polymerase is proportional to the difficulty of the polymerase in copying the repeat: GAA is easier to copy than CAG repeats. Thus, on GAA substrates, some expansions can occur without MSH2-MSH3. The more difficult CAG repeats absolutely require MSH2-MSH3 to create expansion. We discussed this on Page 12, lines 278-280.

5) (Minor point) Can the authors really be sure that the peak labeled CAG18 is really 18 and not 17? The spacing between the peak labeled CAG19 and CAG18 seems a little large.

Yes, we are sure. The capillary electrophoresis is very high resolution, and is the method of choice for distinguishing single nucleotide length differences (see *Tutorial, The chemical educator*, (1996) Vol 1, Springer Verlag (<http://journals.springer-ny.com/chedr>)).

6) Pg. 15, text has been added to the effect that no flap cleavage was seen in the absence of FEN1 (line 347; Fig. 7h, lane 1-3) and later that MSH2-MSH3 alone cannot cut the DNA (line 354; Fig. 7h, lane 7). However, none of those lanes contain both MSH2-MSH3 and Pol β but not FEN1. Without Pol β , there would be no flaps to cut.

According to the reviewer's suggestion, we tested whether any DNA cleavage occurred when pol β and MSH2-MSH3 were present in the absence of FEN1. There are no cleavage products formed in these reactions. This is consistent with the fact that MSH2-MSH3 does not exhibit nuclease activity. We included these results in a new Supplementary Figure 10 and the results are discussed on Page 15, line 347-348.

REVIEWERS' COMMENTS:

Reviewer #1 (Remarks to the Author):

I have no major remarks with respect to my previous positive evaluation of the revised version.

Reviewer #2 (Remarks to the Author):

I have supplied my comments as a PDF file. (below)

(1g) Fig. S5 is unnecessary.

(I assume that the reviewer is referring to Fig. S4).

We agree that Fig. S4 is unnecessary. However, we have exceeded the word limit and have been asked by the editors to reduce the main text. Thus, we have left in Fig. S4, and refer the reader to the legend (Supplementary information, Page 4) and to the methods (Page 25, lines 607-611). However, if the reviewer feels strongly that we should remove it, we will do so. Please let us know.

Since Fig. S4 is lifted directly out of a Zeiss manual (without proper attribution), for what I assume is simply for teaching purposes, simply citing the manual (the link provided is broken), would be more appropriate and there are certainly places in the main text where some words could be deleted without ill-effect. At the very least, the source of the figure and text should be made absolutely clear. However, I will defer to editorial policy on this matter. I have uploaded a copy of the manual in question. The relevant figure is Fig. 3 and the text is located on Pg. 5 and Pg. 8 as shown below:

Figure 3: Selecting specific channels for colocalization analysis in ZEN

Colocalization analysis is performed on a pixel by pixel basis. Every pixel in the image is plotted in the scatter diagram based on its intensity level from each channel. The color in the scatterplot represents the number of pixels that are plotted in that region. In this example, green intensity is shown on the x-axis and red intensity is shown on the y-axis. You can select which channel you want on which axis (See Figure 3 for ZEN and Figure 4 for AIM). For an image with three channels or more, only two channels can be selected for colocalization analysis.

The lower left, unlabeled quadrant in the scatterplot represents pixels that have low intensity levels in both channels (Quadrant 4); these pixels are referred to as background and are not taken into consideration for colocalization analysis. In this example, Quadrants 1 represent pixels that have high green intensities and low red intensities and Quadrant 2 represents pixels that have high red intensities and low green intensities. Quadrant 3 represents pixels with high intensity levels in both green and red. These pixels are considered to be colocalized.

(2) *"In the original manuscript it was stated that "Pol β was unable to traverse the repeat loop on the template generating a large single stranded region that was not seen on a template that did not contain the repeat (Fig. 6 and Supplementary Fig. 7)." The reviewer comments that the data presented did not distinguish a large loop from a series of small loops repeated during strand-slippage. The reviewer did not feel that we answered the questions thoroughly enough, and it was unclear how the as added experiments without dNTPs addressed the question.*

We agree that the data show only that a large single stranded region was formed. Thus, we re-wrote the text to include both possibilities: either a large loop or a series of smaller ones. We agree that the "no dNTP" control demonstrates only that the looped regions required active polymerase (Page 13, 303-312, Page 14, line 313).

Lines 304-306 still states " Surprisingly, in the absence of MSH2-MSH3, S1 cleavage generated a distinct ladder of bands 305 between +53 and +23 below the APE1 incision product (Fig. 6d), indicating that the active pol β induced a large single-stranded region in the template strand (Fig. 6d)."

(2) *With respect to Fig. 1, it is stated that MSH2-MSH3 significantly stimulated Pol β loading. It is unclear to me that the authors have evidence that Pol β loading is improved but rather in the presence of both sets of proteins there is more substrate binding. Without specifically examining the levels of free and bound Pol β using something like antibody supershifts or IP, the authors cannot speak to whether Pol β loading is increased or not.*

We agree with the reviewer about this point. We removed the sentence according to the reviewer's comment.

However, line 257 still states: "Since MSH2-MSH3 enhanced binding of pol β onto the TNR template during BER..."

(5) *Also, the major products of GAA repair are expansions even in the absence of MSH2-MSH3. The authors should discuss why MSH2-MSH3 is essential for expansion in most mouse models of the repeat expansion disorders, yet is not obligatory in vitro at least for GAA.*

Good point. Indeed, we believe our results explain, at least in part, why there are expansions upon loss of MSH2-MSH3 in the FRDA mice. The main point of the figure is that the prominent deletion bands is diminished by addition of MSH2-MSH3, and expansions are favored. The general trend for deletion suppression and expansion promotion on GAA and CAG are similar. However, since expansion is determined by the difference between the nucleotides added by pol β synthesis and those removed by FEN1, the need for MSH2-MSH3 to increase processivity of the polymerase is proportional to the difficulty of the polymerase in copying the repeat: GAA is easier to copy than CAG repeats. Thus, on GAA substrates, some expansions can occur without MSH2-MSH3. The more difficult CAG repeats absolutely require MSH2-MSH3 to create expansion. We discussed this on Page 12, lines 278-280.

This section is still a bit confusing since (GAA)₂₀ showed similar ratios of (GAA)₂₀, (GAA)₂₁ and (GAA)₂₂ products with and without MSH2-MSH3, thus it is unclear that MSH2-MSH3 actually promote expansions of the GAA repeat.

5) (Minor point) Can the authors really be sure that the peak labeled CAG18 is really 18 and not 17? The spacing between the peak labeled CAG19 and CAG18 seems a little large.

Yes, we are sure. The capillary electrophoresis is very high resolution, and is the method of choice for distinguishing single nucleotide length differences (see *Tutorial, The chemical educator*, (1996) Vol 1, Springer Verlag (<http://journals.springer-ny.com/chedr>)).

While I appreciate the reference to the resolution of CE, I am quite familiar with the technique. My concern was whether the repeat numbers you assigned to each peak are the right numbers. In this size range (and as indicated by the migration of the markers), the mobility of the PCR fragments should still be in the linear range. Thus fragments in this size range that differ by one repeat should show very similar separation whether one is referring to 18 vs 19 or 20 vs 21. In addition, it seems to me that the peaks in the last two electrophoretograms shown in 5d corresponding to what is labeled CAG18 do not align. I have included a rough diagram to illustrate what I mean.

Black dotted lines correspond to position of the MW marker
Red dotted lines correspond to the nominally CAG18, CAG19 and CAG20 fragments in the bottom panel (with MSH2-MSH3)

Additional points:

Figs. 3 and S7-9 are not numbered according to the order in which they are cited in the text.

Also (and once again my apologies for not noticing these points in earlier versions of the manuscript):

Line 49: The loss of MSH2 and MSH3 in the FRDA mouse did not attenuate expansions.

Line 80: the authors are unnecessarily cautious. The genetic evidence for a role of MSH2-MSH3 is very solid, and I think it would be hard to argue effectively then that it plays no role in expansion. What is controversial is just how this complex acts to produce expansions.

Line 394, since the authors do not directly examine loop formation, perhaps this sentence could be modified slightly as follows "However, it is possible that MSH2-MSH3 is required if the loop is not preformed"

Author response to reviewer comments

1. (1g) Fig. S5 is unnecessary.

(I assume that the reviewer is referring to Fig. S4).

We agree that Fig. S4 is unnecessary. However, we have exceeded the word limit and have been asked by the editors to reduce the main text. Thus, we have left in Fig. S4, and refer the reader to the legend (Supplementary information, Page 4) and to the methods (Page 25, lines 607-611). However, if the reviewer feels strongly that we should remove it, we will do so. Please let us know.

Since Fig. S4 is lifted directly out of a Zeiss manual (without proper attribution), for what I assume is simply for teaching purposes, simply citing the manual (the link provided is broken), would be more appropriate and there are certainly places in the main text where some words could be deleted without ill-effect. At the very least, the source of the figure and text should be made absolutely clear. However, I will defer to editorial policy on this matter. I have uploaded a copy of the manual in question. The relevant figure is Fig. 3 and the text is located on Pg. 5 and Pg. 8 as shown below: (The image cannot be copied and shown here)

According to the suggestions from the reviewer #2 and the editor, Fig. S4 has been removed in the final version of the manuscript.

2. (2) "In the original manuscript it was stated that "Pol β was unable to traverse the repeat loop on the template generating a large single stranded region that was not seen on a template that did not contain the repeat (Fig. 6 and Supplementary Fig. 7)." The reviewer comments that the data presented did not distinguish a large loop from a series of small loops repeated during strand slippage. The reviewer did not feel that we answered the questions thoroughly enough, and it was unclear how the as added experiments without dNTPs addressed the question.

We agree that the data show only that a large single stranded region was formed. Thus, we rewrote the text to include both possibilities: either a large loop or a series of smaller ones. We agree that the "no dNTP" control demonstrates only that the looped regions required active polymerase (Page 13, 303-312, Page 14, line 313).

Lines 304-306 still states " Surprisingly, in the absence of MSH2-MSH3, S1 cleavage generated a distinct ladder of bands 305 between +53 and +23 below the APE1 incision product (Fig. 6d), indicating that the active pol β induced a large single-stranded region in the template strand (Fig. 6d)."

The statement here is consistent with the results of S1 nuclease cleavage. Thus we decide to keep it in the final version of the manuscript.

3. (2) With respect to Fig. 1, it is stated that MSH2-MSH3 significantly stimulated Pol β loading. It is unclear to me that the authors have evidence that Pol β loading is improved but rather in the presence of both sets of proteins there is more substrate binding. Without specifically examining the levels of free and bound Pol β using something like antibody supershifts or IP, the authors cannot speak to whether Pol β loading is increased or not. We agree with the reviewer about this point. We removed the sentence according to the reviewer's comment.

However, line 257 still states: "Since MSH2-MSH3 enhanced binding of pol β onto the TNR template during BER..."

We appreciate the reviewer's careful reading. The sentence was modified according to the reviewer's comment (Page 11, line 276).

4. (5) Also, the major products of GAA repair are expansions even in the absence of MSH2-MSH3. The authors should discuss why MSH2-MSH3 is essential for expansion in most mouse models of the repeat expansion disorders, yet is not obligatory in vitro at least for

GAA.

Good point. Indeed, we believe our results explain, at least in part, why there are expansions upon loss of MSH2-MSH3 in the FRDA mice. The main point of the figure is that the prominent deletion bands is diminished by addition of MSH2-MSH3, and expansions are favored. The general trend for deletion suppression and expansion promotion on GAA and CAG are similar. However, since expansion is determined by the difference between the nucleotides added by pol β synthesis and those removed by FEN1, the need for MSH2-MSH3 to increase processivity of the polymerase is proportional to the difficulty of the polymerase in copying the repeat: GAA is easier to copy than CAG repeats. Thus, on GAA substrates, some expansions can occur without MSH2-MSH3. The more difficult CAG repeats absolutely require MSH2-MSH3 to create expansion. We discussed this on Page 12, lines 278-280.

This section is still a bit confusing since (GAA)₂₀ showed similar ratios of (GAA)₂₀, (GAA)₂₁ and (GAA)₂₂ products with and without MSH2-MSH3, thus it is unclear that MSH2-MSH3 actually promote expansions of the GAA repeat.

Our results showed that MSH2-MSH3 suppressed deletion of GAA repeats. As described on Page 12, lines 299-301, some GAA repeat expansion occurred without MSH2-MSH3. However, CAG repeat expansion absolutely required MSH2-MSH3. This may result from more efficient synthesis of GAA than CAG repeats by pol β . In addition, to avoid any confusion, we removed “**Fig. 5c**” from the sentence on Page 12, line 294.

5. 5) (Minor point) Can the authors really be sure that the peak labeled CAG₁₈ is really 18 and not 17? The spacing between the peak labeled CAG₁₉ and CAG₁₈ seems a little large. Yes, we are sure. The capillary electrophoresis is very high resolution, and is the method of choice for distinguishing single nucleotide length differences (see Tutorial, The chemical educator, (1996) Vol 1, Springer Verlag (<http://journals.springer-ny.com/chedr>).

While I appreciate the reference to the resolution of CE, I am quite familiar with the technique.

My concern was whether the repeat numbers you assigned to each peak are the right numbers. In this size range (and as indicated by the migration of the markers), the mobility of the PCR fragments should still be in the linear range. Thus fragments in this size range that differ by one repeat should show very similar separation whether one is referring to 18 vs 19 or 20 vs 21. In addition, it seems to me that the peaks in the last two electrophoretograms shown in 5d corresponding to what is labeled CAG₁₈ do not align. I have included a rough diagram to illustrate what I mean. (The image cannot be shown)

We are very appreciative of the reviewer’s careful reading. We went through the original data of DNA fragment analysis carefully again and found that in fact, the length of the product should be (CAG)₁₇ as the reviewer pointed out. We corrected the error in Figure 5.

6. Additional points:

1) Figs. 3 and S7-9 are not numbered according to the order in which they are cited in the text. Also (and once again my apologies for not noticing these points in earlier versions of the manuscript):

The Figs. 3 and S7-9 are arranged according to the order they are cited in the text of the manuscript by removing the sentence “The interaction between them appeared to be direct (**Fig. 3e**). Even in the absence of DNA, the two enzymes were “pulled down” together by immunoprecipitation reactions (**Fig. 3e**); MSH2-MSH3 was present in the immunoprecipitates of pol β , and *vice versa* (**Fig. 3e**).” as well as by renumbering the Supplementary Figures.

2) Line 49: The loss of MSH2 and MSH3 in the FRDA mouse did not attenuate expansions.

We appreciate the reviewer's careful reading. In fact, a previous study by Bourn et al. has shown that the loss of MSH2 can significantly reduce the GAA repeat expansions in the FRDA mice (Bourn, R.L. et al. *PLoS One* **7**, e47085 (2012)). However, we found that the work conducted by Ezzatizadeh, V. et al. (Ezzatizadeh, V. et al. (2014) MutLalpha heterodimers modify the molecular phenotype of Friedreich ataxia. *PLoS One* **9**, e100523) was miscited here. We removed it and reformatted the references (Page 3, line 54).

3) Line 80: the authors are unnecessarily cautious. The genetic evidence for a role of MSH2-MSH3 is very solid, and I think it would be hard to argue effectively then that it plays no role in expansion. What is controversial is just how this complex acts to produce expansions.

We modified the sentence according to the reviewer's comment (Page 4, line 86).

4) Line 394, since the authors do not directly examine loop formation, perhaps this sentence could be modified slightly as follows "However, it is possible that MSH2-MSH3 is required if the loop is not preformed"

We modified the sentence according to the reviewer's suggestion (Page 16, line 423).